# Membrane potential drives the exit from pluripotency and cell fate commitment via calcium and mTOR

Emily Sempou[1], Valentyna Kostiuk[1], Jie Zhu[2], M. Cecilia Guerra[3], Leonid Tyan [1,2], Woong Hwang[1], Elena Camacho-Aguilar [3], Michael J. Caplan[2], David Zenisek [2], Aryeh Warmflash[3], Nick D. L. Owens [4] & Mustafa K. Khokha [1]✉

Transitioning from pluripotency to differentiated cell fates is fundamental to both embryonic development and adult tissue homeostasis. Improving our understanding of this transition would facilitate our ability to manipulate pluripotent cells into tissues for therapeutic use. Here, we show that membrane voltage ($V_m$) regulates the exit from pluripotency and the onset of germ layer differentiation in the embryo, a process that affects both gastrulation and left-right patterning. By examining candidate genes of congenital heart disease and heterotaxy, we identify *KCNH6*, a member of the ether-a-go-go class of potassium channels that hyperpolarizes the $V_m$ and thus limits the activation of voltage gated calcium channels, lowering intracellular calcium. In pluripotent embryonic cells, depletion of *kcnh6* leads to membrane depolarization, elevation of intracellular calcium levels, and the maintenance of a pluripotent state at the expense of differentiation into ectodermal and myogenic lineages. Using high-resolution temporal transcriptome analysis, we identify the gene regulatory networks downstream of membrane depolarization and calcium signaling and discover that inhibition of the mTOR pathway transitions the pluripotent cell to a differentiated fate. By manipulating $V_m$ using a suite of tools, we establish a bioelectric pathway that regulates pluripotency in vertebrates, including human embryonic stem cells.

Action potentials are fundamental to the function of excitable cells, including neurons, cardiomyocytes, and pancreatic cells. They are produced through tightly orchestrated changes in the membrane potential ($V_m$). However, most animal cells, excitable or not, have a resting state $V_m$ (resting membrane potential) that depends on (a) the permeability of the plasma membrane for each ion ($p$ in the Goldman-Hodgkin-Katz (GHK) equation, Fig. 1a, indicating the number of active ion channels), and (b) the driving force for each ion across the plasma

membrane, determined by its electrochemical gradient (e.g., $[K]_i$ vs. $[K]_o$ in GHK equation). Although molecules that influence membrane potential have established roles in excitable tissues, their functions in embryonic or adult "non-excitable" tissues are emerging. For example, ion channels and pumps appear to be crucial for the formation of the left-right (LR) body axis[1]. While the vertebrate body plan may appear symmetrical across the LR axis, some of our internal organs, including the heart and gut, require asymmetry across the LR axis for proper

[1]Pediatric Genomics Discovery Program, Departments of Pediatrics and Genetics, Yale University School of Medicine, 333 Cedar Street, New Haven, CT 06510, USA. [2]Department of Cellular and Molecular Physiology, Yale University School of Medicine, 333 Cedar Street, New Haven, CT 06510, USA. [3]Departments of Biosciences and Bioengineering Rice University, 345 Anderson Biological Labs, Houston, TX 77005, USA. [4]Department of Clinical and Biomedical Sciences, University of Exeter, Barrack Road, Exeter EX2 5DW, UK. ✉e-mail: mustafa.khokha@yale.edu

**Fig. 1 | Membrane potential is important for gastrulation and LR patterning.**
**a** GHK equation for $V_m$; R = gas constant, T = temperature, F = Faraday's constant, $p$ = permeability for each ion, $[X]_o$ = ion concentration outside of cell, $[X]_i$ = ion concentration inside the cell. **b**–**i** Effects of depolarizing treatments (barium chloride and high $K^+$) and *kcnh6* depletion on gastrulation (**b**–**e**; stage 15–17 embryos; arrowheads indicate incomplete blastopore closure) and organ *situs* (**f**–**i**; stage 45 tadpoles; ventral views; arrowheads indicate normal (D) and inverse (L) heart looping). **j** Different stages of barium chloride application (color key in **j** for bar graphs **k** and **l**; green = cleavage stages (stages 0–6 or 0–8); red = gastrulation (stages 8–12); blue = LRO signaling (stages 12–19); orange = early organogenesis (stages 19–30); gray = cleavage through LRO signaling (stages 0–19). *Xenopus* illustrations © Natalya Zahn (2022) from Xenbase (www.xenbase.org RRID:SCR_003280). **k**, **l** Percentages of embryos with incomplete blastopore closure at stage 15 (**k**) and abnormal organ situs at stage 45 (**l**) after treatment with barium or high $K^+$ at different stages (see **j** for color code); *p*-values are in **k**: ($Ba^{2+}$ gastrula vs untreated) = 2.28e−015, (high $K^+$ gastrula vs untreated) = 1.36e−006; in **l**: ($Ba^{2+}$ gastrula vs untreated) = 5.85e−007, (high $K^+$ gastrula vs untreated) = 2.08e−008, ($Ba^{2+}$/LRO vs untreated) = 8.51e−003, and ($Ba^{2+}$ 0–19 vs untreated) = 1.2e−006. All graphs depict mean ± SEM and report total embryo numbers (N) collected over 3 independent experiments for high $K^+$, and over 2 independent experiments for the 47 h barium time course. Key for asterisks: *$p \le 0.05$, **$p \le 0.01$, ****$p \le 0.0001$, ns nonsignificant with $p > 0.05$. Source data are provided as a Source Data file.

and chondrogenesis[12,13] as well as the differentiation of excitable tissues such as muscle cells and neurons[14]. A challenge in the field is connecting changes in $V_m$ to voltage responsive effectors that lead to the gene expression changes that pattern the embryo.

In order to respond, voltage sensitive effector molecules depend on the magnitude in the change of $V_m$. Quantitative $V_m$ measurements in early embryos are rare but were performed in the 1960s and 1970s from 1-cell stage embryos through blastula stages in *Triturus* and *Xenopus* embryos[15,16]. The blastula embryo has completed a series of rapid cell divisions (cleavages), has established germ-layer cell fates (ectoderm, mesoderm, and endoderm), and is poised to begin gastrulation, the process by which cell movements transform the embryo to acquire the adult body plan. Notably, while the $V_m$ at early cleavage stages is depolarized (=more positive $V_m$) ($V_{m@2\text{-}cell} = -19 \pm 10$ mV), it becomes progressively hyperpolarized (=more negative) towards blastula stages (−50 mV)[16]. The implications of this progressive $V_m$ polarization during early development are unclear, as is a mechanism by which $V_m$ could transduce a signal within embryonic cells or act complementary to signals transduced biochemically (i.e., ligand-receptor).

Here we show that the $V_m$ established in the blastula is essential for LR patterning and the exit from pluripotency. Depolarization of $V_m$ using a variety of approaches leads to loss of ectodermal and paraxial mesodermal cell fates due to a persistence of pluripotency in these tissues. Membrane depolarization leads to the opening of voltage-gated calcium channels elevating intracellular calcium and maintaining pluripotency. Using RNA-seq, we find that mTOR signaling is down-stream of $V_m$ and modulates the transition from pluripotency to differentiated cell fate. These results define an electrochemical signaling pathway that acts complementary to biochemical (ligand-receptor) signaling pathways that transition pluripotent embryonic stem cells to differentiated cell fates.

## Results

### $V_m$ of the blastula regulates LR patterning and gastrulation

To address the question of when $V_m$ is critical for embryonic development, we employed barium ions to block $K^+$ channels at different time points of embryonic development, since $K^+$ conductance is paramount for determining $V_m$. Because $K^+$ conductance drives the membrane voltage to a negative (hyperpolarized) potential, blocking $K^+$ channels depolarizes cells. In line with the previous electrophysiological evidence demonstrating that embryos first become polarized at the blastula stage[16], we found that Barium treatment affected embryonic development primarily when embryos were treated from blastula stages through gastrulation rather than at earlier cleavage stages (Fig. 1b, c, f, g, j–l)[17]. Embryonic development was affected in two ways: (1) 37 ± 3% (SEM) of the embryos failed to complete gastrulation (compared to just 4.7 ± 1% in control embryos) (Fig. 1b, c, j, k), and (2) 23 ± 2% that completed gastrulation exhibited misplacement of their organs relative to the left-right axis (compared to just 4 ± 2% in control embryos; Fig. 1f, g, j, l); these included abnormal heart looping to the left, an L-loop (vs a normal D-loop to the right), inverse gut rotation and misplacement of the gall bladder on the left (vs a normal positioning of the gall bladder on the right side of the body axis) (Fig. 1f, g). Because Barium can affect more than just $K^+$ channels, we tested an alternative strategy for achieving membrane depolarization, namely manipulating $V_m$ by increasing extracellular potassium ($[K]_o$ in GHK eq. Fig. 1a). Increasing the extracellular potassium reduces the chemical driving force for potassium to leave the cell and decreases the potassium current which depolarizes the embryo (more positive $V_m$). Incubating embryos in high $K^+$ at blastula/gastrula stages was sufficient to cause a) gastrulation failure in 32 ± 6% of embryos (compared to just 6 ± 1% in controls Fig. 1b, d, k) and b) defective

formation or function. Chemical inhibition or overexpression of ion channels or pumps disrupt the proper alignment of internal organs along the left-right axis and affect global LR patterning[2–7]. Notably, using voltage sensitive dyes, $V_m$ appears to vary across the developing embryo suggesting it could play instructive roles[3,6,8]. There is now a growing field that has implicated $V_m$ in various embryonic contexts including Drosophila wing patterning[8,9], craniofacial morphogenesis[10,11],

**Table 1 | KCNH gene variants identified in patients with CHD**

| Blinded ID | Gene | Phenotype | Allele type | Class | AA change |
|---|---|---|---|---|---|
| 1-09347 | KCNH1 | CTD/Htx | LOF het | splice | . |
| 1-01004 | KCNH1 | CTD/Htx | LOF het | frameshift_deletion | p.G149fs |
| 1-05070 | KCNH3 | LVO | LOF het | stopgain | p.R139X |
| 1-07611 | KCNH3 | CTD | CmpHet | misD/misD | A911V/ S1021P |
| 1-01856 | KCNH3 | LVO | CmpHet | misD misD | A357T/ F542L |
| 1-05499 | KCNH3 | CTD | de novo | misD | p.E859D |
| 1-02696 | KCNH5 | Htx/TGA | de novo | misD | p.N817S |
| 1-05146 | KCNH6 | Htx | LOF het | stopgain | p.S858X |
| 1-07078 | KCNH6 | CTD | LOF het | stopgain | p.E587X |
| 1-02620 | KCNH6 | other | LOF het | stopgain | p.Q487X |
| 1-06077 | KCNH6 | Htx | LOF het | frameshift_insertion | p.S671fs |
| 1-02515 | KCNH6 | Htx | de novo | misD | p.T274M |
| 1-01783 | KCNH7 | LVO | LOF het | stopgain | p.Y1162X |
| 1-12888 | KCNH7 | CTD | LOF het | stopgain | p.E944X |
| 1-06600 | KCNH8 | Htx/TGA | LOF het | frameshift_deletion | p.D782fs |
| 1-06579 | KCNH8 | other | LOF het | stopgain | p.E576X |

*CTD* conotruncal defect, *Htx* heterotaxy, *LVO* left ventricular outflow tract obstruction, *TGA* transposition of the great arteries, *LOF* loss of function, *CmpHet* compound heterozygote, *misD* damaging missense mutation according to previously published criteria.

organ *situs* at later stages in $28 \pm 3\%$ of embryos (compared to just $4 \pm 2\%$ in controls; Fig. 1f, h, l). Thus, our results suggest that establishing proper $V_m$ at blastula stages is essential for both gastrulation and LR patterning, providing context to previous work showing that $V_m$ varies during embryonic development by becoming steadily more polarized from egg to blastula.

### KCNH6 is essential for LR patterning and gastrulation

Recent studies in patients with congenital heart disease identified a number of variants in KCNH ether-a-go-go (EAG) potassium channels (Table 1)[18] as candidate disease genes. Many of these patients had heterotaxy, a disorder of LR development that can have a significant impact on the structure and function of the heart that can be life-threatening. While multiple ions can affect membrane potential, the flow of potassium down its electrochemical gradient ($K^+_{in} \gg K^+_{out}$) has the largest impact on $V_m$ because in most cell types cell membranes are most permeable to potassium. Since KCNH6 was the most common family member in a total of five patients with heterotaxy (Table 1), we began our studies by examining the CHD/Htx candidate gene, KCNH6.

In *Xenopus*, we found *kcnh6* to be expressed in the prospective ectoderm and dorsal/paraxial mesoderm at gastrulation onset, suggesting that it could play a role during gastrulation (Fig. 2a–h). Additionally, high temporal resolution RNA-seq shows that the increase in *kcnh6* transcripts parallels the trend of $V_m$ polarization in the frog embryo (Fig. 2i)[16,19]. We thus tested a role for *kcnh6* in early embryonic development, and in determining $V_m$ specifically. Two F0 CRISPRs independently targeting two different exons in *kcnh6* as well as a translation blocking morpholino oligo (MO) recapitulated the morphological defects observed in embryos depolarized with barium and high $K^+$, including: (a) inability to complete gastrulation (Fig. 3a) (CRex3: $27 \pm 4\%$; CRex4: $29 \pm 5\%$ and MO:$31 \pm 3\%$ vs only $4 \pm 1\%$ in controls), and (b) abnormal organ *situs* (Supplementary Fig. 1a–h) (CRex3: $43 \pm 12\%$; CRex4: $19 \pm 1\%$ and MO:$91 \pm 7\%$ vs only $5 \pm 3\%$ in controls). Importantly, we established specificity of our depletion studies by the following criteria: (1) phenocopy with two non-overlapping sgRNAs via F0 CRISPR and one translation blocking MO (Fig. 3a and Supplementary Fig. 1a–h), (2) rescue of the MO phenotype with human *KCNH6* mRNA (Fig. 3a), and (3) detection of gene editing of the *kcnh6* locus by PCR amplification and Inference of CRISPR Edits (Supplementary Fig. 2)[20]. Moreover, treatment of blastula/gastrula embryos (stage 8 to

stage 12) with Ergtoxin, a scorpion peptide that specifically acts as a pore blocker of the KCNH channel family[21], also led to identical gastrulation and LR defects (Fig. 3a and Supplementary Fig. 1f–h). These results indicate that Kcnh channels, and specifically Kcnh6, contribute to gastrulation and LR development, consistent with their identification in patients with Htx/CHD.

### $V_m$, rather than KCNH6 per se, is essential for gastrulation

Depletion/inhibition of potassium channels or elevation of extracellular $K^+$ should lead to membrane depolarization. Thus, we reasoned that the inverse condition, namely hyperpolarizing by reducing extracellular $K^+$, should rescue *kcnh6*-depleted embryos. Lowering extracellular $K^+$ ($[K]_o$ in GHK eq. Fig. 1a) increases the outward driving force for flow of $K^+$, provided that other $K^+$ channels are present. Indeed, lowering extracellular $K^+$ rescues the gastrulation defect in *kcnh6*-depleted embryos (CRISPR and MO) (Fig. 3a and Supplementary Fig. 3a). To test the significance of $K^+$ conductance independently of a specific $K^+$ channel, we employed valinomycin, a $K^+$ selective ionophore that inserts itself into the plasma membrane and mimics the ability of a $K^+$ channel to passively conduct $K^+$ down its electrochemical gradient. Application of valinomycin also rescued the gastrulation defect in *kcnh6*-depleted embryos ($30 \pm 1\%$ gastrulation defect in *kcnh6* CR/DMSO- vs $16 \pm 7\%$ in *kcnh6* CR/Valinomycin-treated embryos) emphasizing the importance of $K^+$ flux rather than a specific need for Kcnh6 itself or an alternative role of Kcnh6 in cell signaling (Fig. 3a and Supplementary Fig. 3a). Finally, to differentiate between membrane potential and an alternative role for $K^+$ flux across the plasma membrane (e.g., cell volume regulation), as well as to unlink membrane potential from a specific conductance (i.e., potassium), we hyperpolarized by reducing extracellular $Na^+$ and replacing it with equimolar choline. Choline has equivalent cationic charge to $Na^+$ but cannot pass through channels and therefore does not influence the $V_m$. Importantly, replacement of $Na^+$ with equimolar choline does not affect the osmotic properties of the medium, ensuring that external $Na^+$ depletion will only act on $V_m$ and not on cell volume[22]. To determine the amount of sodium to replace with choline, we measured $V_m$ using intracellular electrodes in the animal pole in the context of high $K^+$ and rescue by choline substitution of sodium. Embryos exposed to high $K^+$ were depolarized ($V_m = -20 \pm 4$ mV) while embryos treated with high $K^+$ and ½ sodium replacement with choline were repolarized to roughly

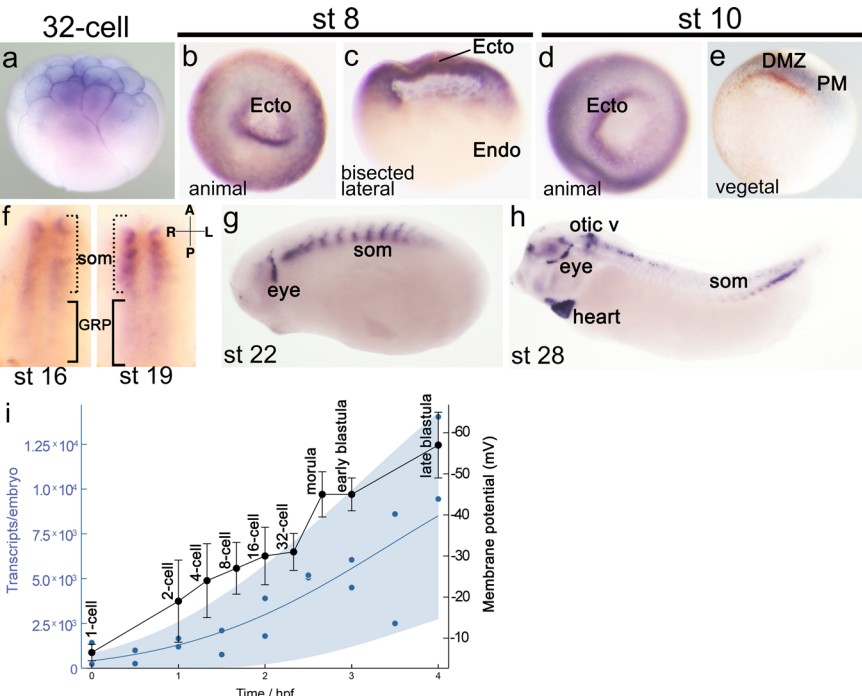

**Fig. 2 | kcnh6 expression analysis for *kcnh6* during *Xenopus* development.**
**a**–**h** *kcnh6* transcripts were detected by a full-length antisense probe via WMISH. Embryos and tissues displayed are in the following orientations: **a** animal pole to the top, **b** animal pole view **c** bisected with animal pole to the top, **d** animal pole view, **e** vegetal view and dorsal to the top, **f** gastrocoel roof plate with anterior to the top and vegetal view, **g**, **h** lateral view with anterior to the left and dorsal to the top; Ecto prospective ectoderm, Endo prospective endoderm, DMZ dorsal marginal zone (mesoderm), PM paraxial mesoderm, som somites, GRP gastrocoel roof plate. Representative images from *N* = 60 embryos (per developmental stage) over 3 independent experiments. **i** kcnh6 transcripts[19] (blue) and $V_m$[16] (black) plotted during early *Xenopus* development. Blue dots represent mean transcript levels by temporal resolution RNA-seq, while blue shaded region marks Gaussian process 95% confidence interval of the data; *n* = 2 biological replicate time courses in original study. Membrane potential is presented as mean ± SD from at least *n* = 14 embryos in original study.

the normal membrane potential ($V_m = -41 \pm 4$ mV) (Supplementary Fig. 3b). Therefore, we tested ½ sodium replacement with choline on embryos and assayed for gastrulation which led to a remarkable rescue in *kcnh6*-depleted embryos (Fig. 3a). These data suggest that $V_m$, which is determined by the conductance of $K^+$ through Kcnh6 and influenced by other $K^+$ and $Na^+$ channels, is key to gastrulation.

Finally, we sought to measure the change in $V_m$ when *kcnh6* is depleted. Using intracellular electrodes in the animal pole of *kcnh6* MO vs control MO-injected embryos at gastrulation onset, we recorded a $V_m$ of $-24 \pm 1.9$ mV in *kcnh6* MO vs $-44 \pm 2.1$ mV in control MO embryos (Fig. 3b, c). Thus, Kcnh6 contributes -20 mV to the cell's negative resting potential, and embryos lacking *kcnh6* are abnormally depolarized compared to their control counterparts.

### Depolarized $V_m$ increases calcium levels

We then asked how $V_m$ is transduced into a signal that affects embryonic development. There are a limited number of voltage responsive elements in a cell. We reasoned that depolarization ($V_m = -24$ mV) in *kcnh6*-depleted embryos could aberrantly activate voltage-gated $Ca^{2+}$ channels (VGCCs), which facilitate inward $Ca^{2+}$ flux[23]. L-type VGCCs are present in the prospective ectoderm and dorsal mesoderm and can induce potent intracellular $Ca^{2+}$ increases that can alter germ-layer patterning[24,25], yet upstream regulators of these calcium channels remain elusive. Interestingly, intracellular $Ca^{2+}$ is elevated after fertilization and during early cleavage stages but declines as the embryo approaches gastrulation[26] concomitant with the onset of membrane polarization. We argued that, if VGCCs are aberrantly activated due to an abnormally depolarized $V_m$, we should be able to detect changes in intracellular $Ca^{2+}$ levels. To assess this, we microinjected the calcium indicator GCaMP6[27] mRNA together with mCherry mRNA (to enable ratiometric analysis) into control MO or *kcnh6* MO embryos and

performed calcium imaging in animal cells of early gastrula embryos. Within the animal pole of stage 10 control MO-injected embryos, we observed multiple intracellular calcium increases, signified by a pulse-like appearance of GCaMP6 fluorescence in isolated cells, which then propagated to adjacent cells (Supplementary Movie 1). These increases are well documented in *Xenopus* stage 8–12 gastrulae, i.e., last a few seconds, in which they spread to adjacent cells and then extinguish, are VGCC dependent and may contribute to neural induction[25,28]. We confirmed the existence of $Ca^{2+}$ transients at stage 10 by performing 20 s time lapse recordings, and additionally observed that they are of low intensity and typically do not simultaneously affect more than $16 \pm 10\%$ of the total animal pole area (Supplementary Movie 1 and Fig. 3d–f). Interestingly, the same transients were dramatically increased in stage 10 *kcnh6* MO embryos both in intensity and area (Supplementary Movie 2 and Fig. 3d–f), affecting $71 \pm 11\%$ of the animal pole on average, with most embryos displaying simultaneous calcium increases in >90% of the animal pole. Thus, *kcnh6* contributes to a hyperpolarized $V_m$ and is key for suppressing calcium levels at gastrulation onset, a signal that may facilitate correct gastrulation.

### Depolarization affects paraxial mesoderm and ectoderm

For gastrulation to proceed normally, two steps are critical: first, the germ layers of the blastula embryo (ectoderm, mesoderm, and endoderm) must be patterned correctly and second, the embryo must undergo the cellular rearrangements that drive morphogenesis. Calcium has been previously identified to play a role in morphogenesis cell behaviors during gastrulation[29,30]. Alternatively, calcium may play a role in patterning the mesoderm that also drives gastrulation cell movements. Patterning precedes morphogenesis, and morphogenesis can fail as a result of abnormal patterning. We, therefore, first examined if patterning is disrupted in $V_m$-depolarized embryos via marker

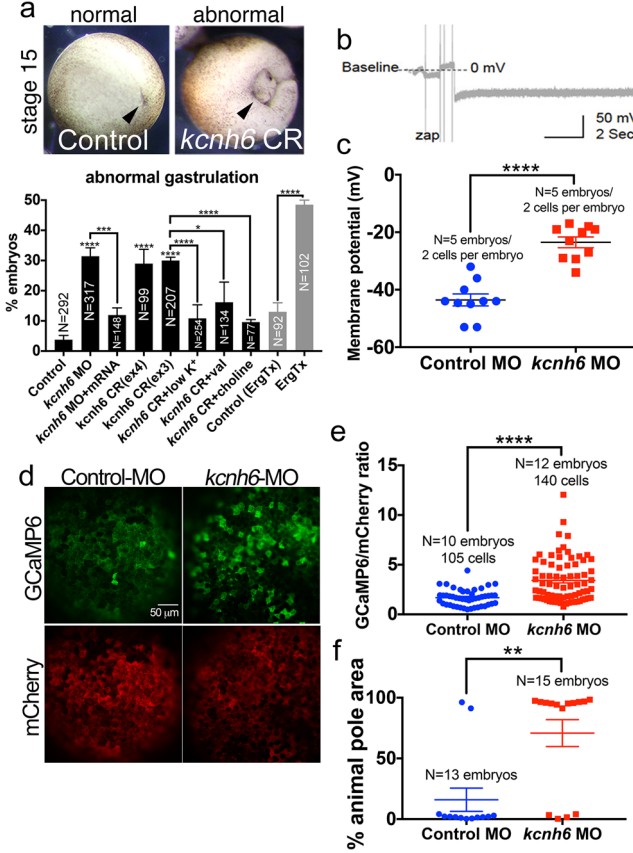

**Fig. 3 | Membrane potential is important for gastrulation and regulates calcium levels at gastrulation onset. a** Percentages of embryos with abnormal gastrulation after depletion of *kcnh6* (MO, CRISPR) or Kcnh channel blockade with Ergtoxin, and rescue of *kcnh6* depletion with medium conditions that hyperpolarize the $V_m$ (low $K^+$, val valinomycin, sodium substitution with choline; treatments performed stages 8–12). Above: examples of embryos scored for the graph; posterior views (dorsal to the top) of stage 15 embryos after successful (control) or unsuccessful (*kcnh6* CRISPR = CR) gastrulation; arrowhead points to blastopore closure. Graph reports mean ± SEM; total embryo numbers (N) in the graph are from 3 independent experiments (except for Ergtoxin: 2 independent experiments with devitellinized embryos); *p*-values are (*kcnh6*MO vs Control MO) = 8.66e-010, (*kcnh6*MO+mRNA vs *kcnh6*MO) = 2.59e-004, (*kcnh6*CRex4 vs Control) = 7.19e-018, (*kcnh6*CRex3 vs Control) = 5.79e-022, (*kcnh6*CR+low $K^+$ vs *kcnh6*CR) = 1.64e-005, (*kcnh6*CR+val vs *kcnh6*CR+DMSO) = 1.22e-002, (*kcnh6*CR +choline vs *kcnh6*CR) = 2.34e-006, (ErgTx vs Control) = 6.57e-010; two-sided Fisher's exact test. **b** Representative intracellular recording in the prospective ectoderm of a control stage 10 embryo; $V_m$ is measured relative to the medium (baseline); the dip in membrane potential indicates the electrode breaking into the cell. **c** The $V_m$ as measured by intercellular recordings in the prospective ectoderm of stage 10 Control MO and *kcnh6* MO-injected embryos; graph reports mean ± SEM; *p*-value (*kcnh6*MO vs Control MO) is 1.07e-006 (unpaired two-tailed student's *t*-test); each data point represents one cell; data from 10 cells/5 embryos/3 independent experiments. **d** Live animal pole images of GCaMP6/ mCherry at stage 10. **e** Quantification of GCaMP6 fluorescence intensity normalized to mCherry in mCherry+ cells; graph shows mean ± SEM; data points represent single cells; data from N cells (in graph)/10 embryos/3 independent experiments; *p* = 1.14e-006; unpaired two-tailed student's *t*-test. **f** Maximum area undergoing simultaneous $Ca^{2+}$ transients within a 20 s time lapse recording as a percentage of total animal pole area; the animal poles of 13 Control MO and 15 Kcnh6 MO embryos were recorded over 3 independent experiments; *p* = 1.04e-003; unpaired two-tailed student's *t*-test. Key for asterisks: *$p ≤ 0.05$, **$p ≤ 0.01$, ***$p ≤ 0.001$, ****$p ≤ 0.0001$, ns nonsignificant with $p > 0.05$. Source data are provided as a Source Data file.

gene expression. Since the mesoderm is critical for gastrulation movements, we began with this germ layer. Markers of the dorsal (*gsc*, *nodal3*) and ventral mesoderm (*vent2*) appeared unaffected in *kcnh6*-depleted, barium and high $K^+$ depolarized embryos (Supplementary Fig. 4e–j); however, the paraxial mesoderm fates appeared lost as marked by *myoD*, *myf5*, and *tbxt* (*brachyury*, *xbra*) (Fig. 4a–d and Supplementary Fig. 4a–d). In fact, absent patterning of paraxial mesoderm by *myoD* persisted into the Left-Right Organizer (Supplementary Fig. 5), a transient structure formed at the end of gastrulation where cilia driven flow is thought to pattern the LR axis[31,32]. In the LRO, *dand5* (*coco*) is normally expressed in the paraxial mesoderm symmetrically until cilia driven flow suppresses *dand5* expression on the left[33,34]. However, consistent with a mispatterning of the paraxial mesoderm in the LRO, *dand5* was also absent even before the occurrence of cilia driven flow (Supplementary Fig. 1i, j). A disruption in the LRO is further supported by defective *pitx2c* expression in the left lateral plate mesoderm at later stages (Supplementary Fig. 1k, l). Therefore, in *kcnh6*-depleted embryos, the paraxial LRO is mispatterned, and a defect in this tissue can be detected already at the onset of gastrulation.

When certain biochemical signaling factors are depleted, loss of one-cell fate (e.g., paraxial mesoderm) is often concomitant with gain of another cell fate[35–37]. Since the dorsal or ventral mesoderm appeared unaffected (Supplementary Fig. 4e–j), we considered that the ectoderm or endoderm might be expanded into the mesodermal area of embryos with abnormally depolarized $V_m$[38,39]. Interestingly, while the endoderm (*vegT*) and its border to the mesoderm (*mixer*) seemed unaffected (Fig. 4g–j), ectodermal fates (*ectodermin* and *foxI1a*) were lost similar to the paraxial mesoderm (Fig. 4e, f and Supplementary Fig. 4k, l). In fact, depletion of *ectodermin* (*trim33*) leads to developmental arrest midway through gastrulation[39], which corresponds to the arrested phenotype in a portion of depolarized embryos. These results indicate that $V_m$ has an effect on germ-layer differentiation, and specifically paraxial mesoderm and ectoderm, at gastrulation onset (Fig. 4a–j).

## Cacna1c responds to $V_m$ depolarization

In depolarized embryos, we have established (1) changes in cell fate and (2) elevated intracellular calcium levels, so we next tested if these aberrant cell fates are dependent on voltage-gated calcium channels. To determine the specific embryonic VGCCs downstream of $V_m$, we reviewed our available high temporal resolution RNA-Seq data[19]. *Xenopus* contains detectable transcripts of L- and T-type VGCCs between the 1-cell and gastrula stages, while other VGCC types (N-, R-, and P/Q) are less abundant. L-type VGCCs become activated at $V_m > -40$ mV (and then inactivated at $V_m > 10$ mV) and have been previously implicated in gastrula patterning[23,24,29] while T-type channels become inactivated at $V_m > -60$ mV and would be inactive both at physiological $V_m$ (~−50 mV) and at more depolarized potentials. Therefore, we tested the L-type VGCC blocker nifedipine. This significantly ameliorated both *ectodermin* (ectoderm) and *myf5* (paraxial mesoderm) expression losses in *kcn6* knockdown embryos (Fig. 5a–d). Specifically, *myf5* was lost only in 16 ± 1% and 18 ± 5% of nifedipine-treated *kcnh6* CR and MO embryos (vs 38 ± 5% and 35 ± 5% in *kcnh6*CR or MO embryos treated only with DMSO, Fig. 5b, d). Similarly, absent *ectodermin* was only observed in 16 ± 7% and 17 ± 3% of nifedipine-treated *kcnh6* CR and MO embryos (vs 48 ± 2% and 51 ± 6% in *kcnh6* CR or MO embryos treated with DMSO, Fig. 5a, c). Of the VGCCs identified in our RNA-seq data at blastula/gastrula stages, two genes *cacna1c* (Cav1.2; L-type) and *cacna1g* (Cav3.1; T-type) encode alpha (pore-forming) channel subunits, which are indispensable for channel function. Co-depletion of *cacna1c* in *kcnh6*-depleted embryos rescued expression of *ectodermin* and *myf5*, while co-depleting *cacna1g* (Fig. 5a–d) resulted in no rescue. Thus, Kcnh6, which sets a negative $V_m$, is essential to limit the

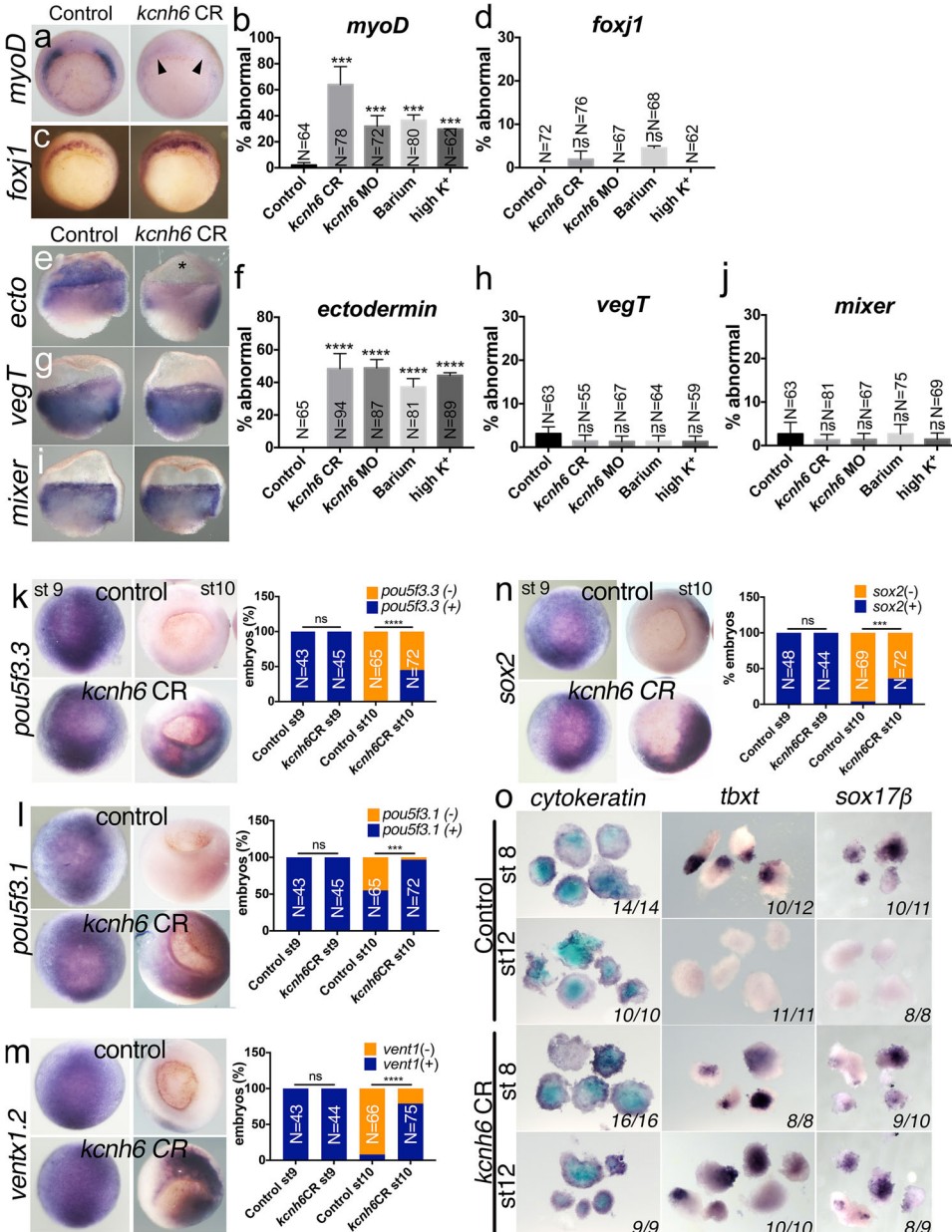

**Fig. 4 | Membrane potential affects early gastrula patterning and pluripotency.**
**a–j** WMISH for germ-layer markers in early gastrula embryos (stage 10; **a, c**: vegetal views with dorsal to the top; **e, g**, and **i**: lateral view of bisected embryos, dorsal to right). Markers are for paraxial mesoderm (*myoD*), superficial dorsal mesoderm (*foxj1*), ectoderm (*ectodermin = ecto*), endoderm (*vegT, mixer*). Graphs (**b, d**) and (**f–j**) depict percentages of embryos with absent or strongly reduced expression of these markers; presented are mean ± SEM; total embryo numbers (N) are from 3 independent experiments; *p*-values (vs Control) are in **b**: (CR vs Control) = 1.61e-006, (MO vs Control) = 4.99e-003, (Ba²⁺ vs Control) = 4.45e-003, (high K⁺ vs Control) = 7.56e-004, and in **f**: (CR vs Control) = 6.81e-013, (MO vs Control) = 2.7e-013, (Ba²⁺ vs Control) = 1.58e-009, (high K⁺ vs Control) = 6.61e-012; ns (nonsignificant) for *p* > 0.05; two-sided Fisher's exact test. **k–n** WMISH for *Xenopus* pluripotency

genes *pou5f3.3, pou5f3.1, ventx1.2*, and *sox2* at stages 9 and 10 (animal pole views). Graphs (**k–n**) depict mean percentages of embryos with present (+/blue) or absent (−/orange) gene expression; total embryo numbers (N) are from 3 independent experiments; p-values (vs Control) are in **k**: *p* = 2.01e-008, in **l**: *p* = 1.52e-010, in **m**: *p* = 6.73e-018 and in **n**: *p* = 1.85e-006; ns (nonsignificant) for *p* > 0.05. Key for asterisks in all graphs: *$p \leq 0.05$, ***$p \leq 0.001$, ****$p \leq 0.0001$, ns nonsignificant.
**o** Differentiation potential of animal caps excised at stage 8 or 12 and treated with no activin (differentiation into epidermis marked by cytokeratin), low activin (differentiation into mesoderm marked by tbxt) and high activin (differentiation into endoderm marked by sox17β). In each image, the numbers on the bottom right report caps with the indicated phenotype vs total number of caps analyzed over 2 independent experiments. Source data are provided as a Source Data file.

activation of L-type VGCCs and specifically Cacna1c, a critical step for ectodermal and paraxial mesodermal differentiation.

**Vₘ depolarization maintains pluripotency in gastrula embryos**
The loss of some cell fates (ectoderm and paraxial mesoderm) without a concomitant expansion of other cell fates was puzzling given that most biochemical signaling factors (Wnt, BMP, Nodal) generally balance different cell fates in the early embryo. We speculated that these

unspecified cells may simply lack the ability to assume *any* cell fate because they remain pluripotent abnormally. To test this hypothesis, we examined markers of pluripotency *OCT4*, *NANOG*, and *SOX2*. In *Xenopus*, there are three *OCT4* homologs (*pou5f3.1, 2,* and *3, formerly oct91, oct25,* and *oct60*)[40], and the *ventx1.2/2.2* factors, which have overlapping functions in maintaining differentiation competence and are thought to be structurally and functionally equivalent to mammalian Nanog[41]. *Sox2*, a core pluripotency factor in mammals, is highly

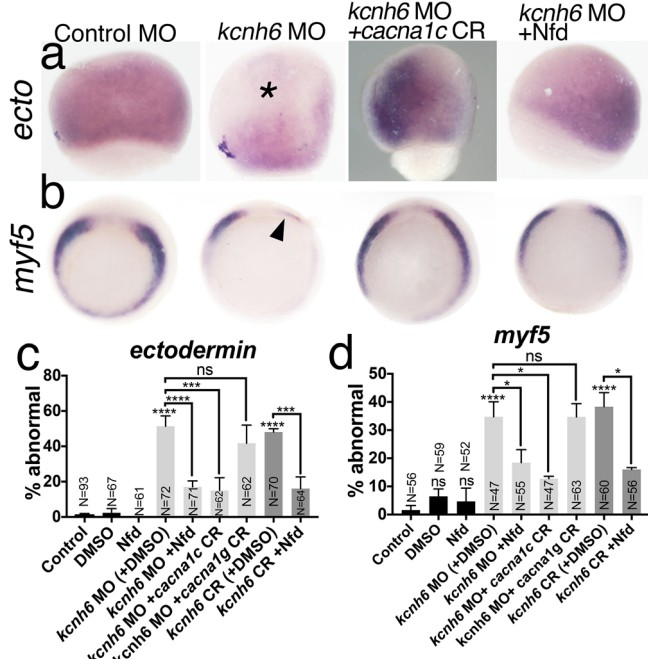

**Fig. 5 | The role of VGCCs in germlayer differentiation. a** WMISH for *ectodermin*; lateral views with the animal pole to the top; asterisk marks the animal pole with loss of *ectodermin* expression; embryos with absent expression are quantified in **c**; CR CRISPR, Nfd nifedipine. **b** WMISH for *myf5*: vegetal views with dorsal to the top; arrowhead marks loss of expression; embryos with abnormal expression are quantified in **d**. **c**, **d** Percentages of embryos with abnormal *ectodermin* (**c**) and *myf5* (**d**) expression. Graphs depict mean ± SEM; *p*-values are in **c**: (*kcnh6*MO+DMSO vs DMSO) = 2.88e-006, (*kcnh6*MO+Nfd vs *kcnh6*MO+DMSO) = 4.59e-002, (*kcnh6* +*cacna1c*CR vs *kcnh6*MO) = 1.53e-002, (*kcnh6*MO+*cacna1g*CR vs *kcnh6*MO) > 9.99 e-001 (ns), (*kcnh6*CR+DMSO vs DMSO) = 3.50e-007, (*kcnh6*CR+Nfd vs *kcnh6*CR +DMSO) = 1.20e-002; in **d**: *p*-values are (*kcnh6*MO+DMSO vs DMSO) = 4.07e-015, (*kcnh6*MO+Nfd vs *kcnh6*MO+DMSO) = 8.92e-005, (*kcnh6*MO+*cacna1c*CR vs *kcnh6*MO) = 4.78e-005, (*kcnh6*MO+*cacna1g*CR vs *kcnh6*MO) = 2.24e-001 (ns), (*kcnh6*CR+DMSO) = 2.04e-014, (*kcnh6*CR+Nfd) = 1.45e-003; two-sided Fisher's exact test; total embryo numbers (N) in the graphs are from at least 2 independent experiments; Key for asterisks: *$p \leq 0.05$, ***$p \leq 0.001$, ****$p \leq 0.0001$, ns non-significant for $p > 0.05$. Source data are provided as a Source Data file.

conserved in amphibians and also expressed at high levels prior to lineage commitment throughout the *Xenopus* blastula[42,43]. We examined the prospective ectoderm of embryos, which is best characterized in its pluripotent properties, and confirmed that *pou5f3.1, pou5f3.3, sox2,* and *ventx1.2* are robustly expressed at stage 9 prior to lineage commitment, but their transcripts are sharply reduced by stage 10 in control embryos (Fig. 4k–n). In contrast, *kcnh6* CR embryos retain robust expression of these factors well beyond stage 9 and into stage 10, a prolonged expression compared to wildtype embryos (Fig. 4k–n). This, in turn, is not due to a general delay in development, since *kcnh6* CR embryos were staged according to the physical progression of gastrulation, i.e. presence of blastopore lip. Moreover, in *kcnh6*-depleted late gastrula embryos, abnormal maintenance of *pou5f3.3* and *ventx1.2* can be abolished by incubating the embryos in L-type VGCC blocker nifedipine (Supplementary Fig. 6b–d). These results suggest that *kcnh6* is upstream of $V_m$ and VGCCs in promoting the exit from pluripotency, which takes place as gastrulation proceeds.

Based on this result, we sought to test the pluripotency of these *kcnh6*-depleted embryos. In the blastula (stage 9), the prospective ectoderm or "animal cap" contains cells that when explanted will differentiate into epidermis (Fig. 4o). Importantly, when stage 8–9 explanted animal cap cells are treated with activin, they can be differentiated into mesodermal and endodermal cell fates indicating that

they are pluripotent (Fig. 4o), an assay used for decades to test the activity of a host of differentiation factors. However, towards the end of gastrulation at stage 12, these animal cap cells are no longer pluripotent and when explanted will only differentiate into epidermis, even when stimulated with activin[43,44] (Fig. 4o). Using this animal cap assay, we sought to test the role of *kcnh6* in determining pluripotency. We explanted stage 9 and stage 12 animal caps and assayed differentiation under three conditions: (1) no activin to examine spontaneous differentiation into epidermis (marked by *cytokeratin*), (2) low activin to stimulate differentiation into mesoderm (*tbxt*), and (3) high activin to stimulate differentiation into endoderm (*sox17β*). Both control and *kcnh6* CR animal caps explanted from stage 9 embryos were capable of differentiating into all three germ layers, indicating full differentiation potential even when *kcnh6* is depleted (Fig. 4o). On the other hand, as expected, animal caps explanted from stage 12 control embryos differentiated into epidermal fate but not into meso- or endoderm despite activin administration (Fig. 4o). Strikingly, stage 12 animal caps explanted concurrently from *kcnh6*-depleted embryos were able to differentiate into cell fates of all three germ layers with activin administration (Fig. 4o). From these and the previous experiments, we conclude that $V_m$ polarization via *kcnh6* enables the exit from pluripotency.

## $V_m$ limits mTOR to allow for exit from pluripotency

Our findings indicate that a polarized $V_m$ limits voltage-gated calcium channels and intracellular calcium, a process that reduces the expression of pluripotency genes as germ-layer differentiation initiates. A critical question is what are the signaling pathways invoked when $V_m$ is depolarized or intracellular calcium is elevated. To address this question in an unbiased manner, we temporally profiled gene expression via RNA-Seq in control and high $K^+$ depolarized embryos by collecting embryos every 30 min from pre- to post-gastrula stages (stages 8–12; Supplementary Fig. 7a–d). We identified genes exhibiting temporal differential expression employing a Gaussian Process framework[19]; this determines genes whose expression trajectory differs between control and high $K^+$ embryos over the time course. We found 4043 genes on average activated in high $K^+$ over the time course, and 1101 genes on average repressed (Supplementary Fig. 7e). We further used k-means clustering to subdivide these into 8 clusters, 4 activated (A1–4, Fig. 6a, b) and 4 repressed (R1-4, Supplementary Fig. 7f, g). In both cases, the clustering segregated genes showing dysregulated gene expression prior to gastrulation (Clusters 1, 2) and during gastrulation (Clusters 3,4) (Fig. 6a, Supplementary Fig. 7f, g). To assess the composition of these clusters, we performed gene set enrichment using Enrichr[45]. Comparing activated clusters to repressed clusters over seven different annotated gene set libraries, we found 1279 terms significantly associated with at least one of the 4 activated clusters, but only 44 terms significantly associated with repressed clusters. Therefore, given the total number of genes and associated terms, we focused our attention on the analysis of the activated genes.

We found a hierarchy of gene set enrichments from early to late in our time course, reflecting the changing response in the transcriptome (Fig. 6c). Notably, we found enrichment for the mTOR signaling pathway activated early with members including *rictor, depdc5, pik3cb, stk11, atg13* in cluster A1 (Fig. 6e) and *akt1s1, gsk3b, lamtor1/2* in cluster A2. The enrichment for mTOR members continues to span all activated clusters (FDR < $3 \times 10^{-5}$ over all activated clusters combined). This enrichment is accompanied by pathways associated with mTOR, including autophagy, ubiquitin transferase activity, and ER response (Fig. 6c). The intermediate clusters A2 and A3 show the most prominent ER response enrichment together with significant upregulation of spliceosome machinery (including 12 snRNPs and 5 SRSF family members). In contrast, in later clusters, we find enrichments for terms explaining the sustained pluripotency and germ-layer defects, including pluripotency (*pou5f3.1, pou5f3.2, ventx1.2*) and WNT signaling (*fzd7,*

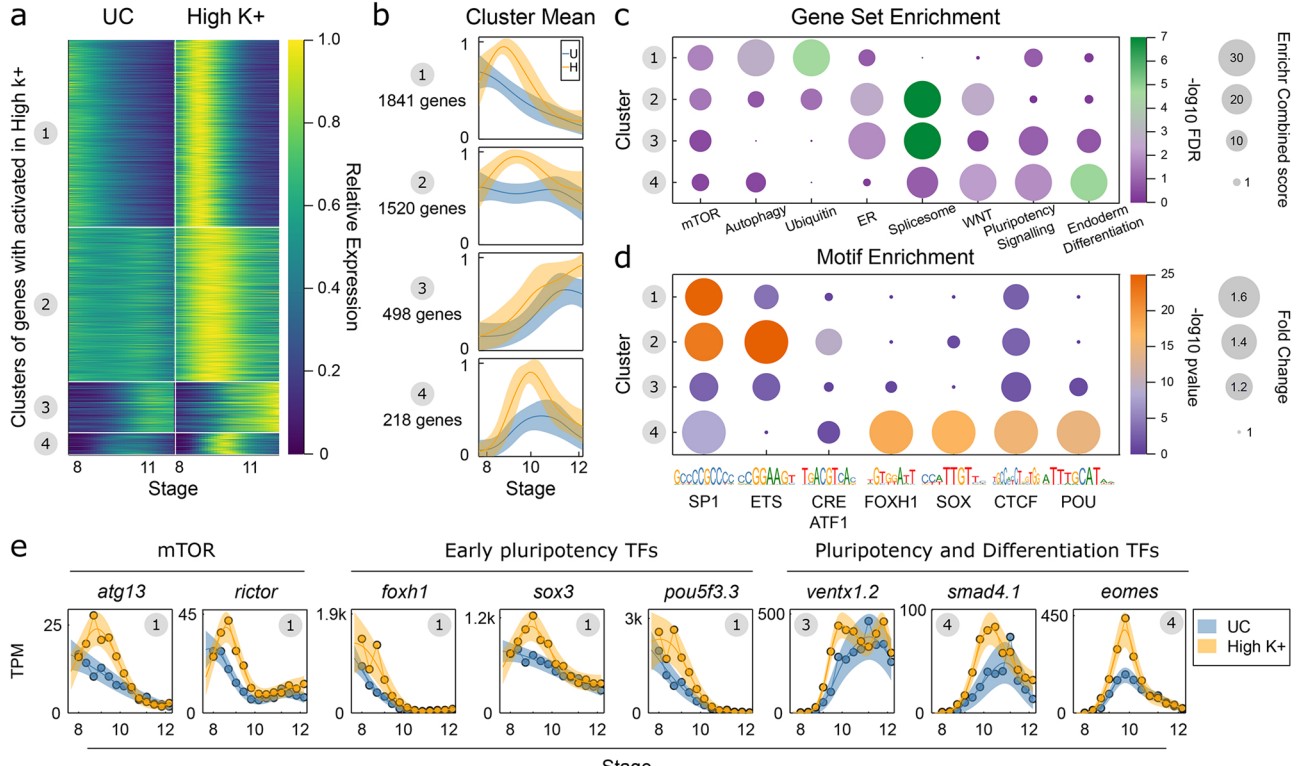

**Fig. 6 | High-resolution temporal RNA-seq identifies mTOR and Ca²⁺ Gene Regulatory Network. a, b** Summary of activated gene clusters by **a** heatmap and **b** cluster average. Data is Gaussian process median for each gene normalized by maximal value, shaded region in **b** is ±1 SD for each cluster. UC untreated control; High K⁺ (=depolarizing conditions). **c** Bubble plot for selection of gene set enrichment terms, calculated with Enrichr, see Methods for definition of terms, and Supplementary Data 1 for full set of enrichments. Bubble size reflects Enrichr Combined score and color indicates −log10 FDR. **d** Bubble plot enrichment of TF motifs in 500 bp upstream of cluster promoters, see also Supplementary Fig. 7g. Bubble size reflects fold change over background and color is −log10 Hypergeometric right tail *p*-value for enrichment. **e** Expression of exemplar genes in control and high K+. Central line and shaded region are transformed Gaussian process median and 95% CI. Circle in top right hand corner gives cluster number. UC (=untreated control), High K⁺ (=depolarizing conditions). Data analysis performed from N = 13 samples/each 10 embryos over one biological replicate time course.

*wnt8a, tcf7l1*) (Fig. 6c, e). Key members of the *Xenopus* pluripotency network are also found activated early, in cluster A1 including *foxh1*, *sox3*, and *pou5f3.3* (Fig. 6c, e).

To build the underlying gene regulatory networks, we examined transcription factor motif enrichment in the promoters of each of these activated gene clusters. Mirroring the gene set enrichments, we found motif enrichments segregated between early (A1,2) and later (A3,4) gene clusters (Fig. 6d, Supplementary Fig. 7h). We find strong enrichment for SP1, ETS, YY1, RFX, and CRE/ATF1 motifs driving gene expression changes in clusters A1 and A2. Interestingly, both ETS and CRE motifs are bound by factors responsive to calcium. Calcium induced phosphorylation of ETS1 inhibits binding activity[46], and CRE elements are bound by calcium responsive family members, CREB1, CREM, and ATF1[47–49]. In our data, each of these factors are expressed at high levels at stage 8, and then are gradually downregulated to a minimum at stage 12. Of note, their mRNAs are not upregulated in high K⁺ conditions (Supplementary Fig. 7i). This suggests that the activity of these factors is post-translationally modified in depolarized embryos experiencing high Ca²⁺ to drive gene expression changes. In the case of ETS1, this factor may act to repress gene expression in normal germ-layer resolution and this repression is removed in high Ca²⁺ embryos. Supportive of these factors driving gene expression, we find a large intersection between activated gene clusters A1 and A2 and genes found in proximity to publicly available ETS1, CREB1 and CREM binding sites (Supplementary Fig. 7j); in the case of activated cluster A2 this remarkable enrichment accounts for 794/1520 (52.2%) of genes found in proximity to one of ETS1, CREB1, and CREM (FDR < 10⁻²⁶,10⁻²³,10⁻³⁵, respectively, Fisher Exact Test). Therefore, we find that initial

transcriptional responses captured by activated gene clusters A1 and A2 appear to be largely driven directly by calcium response. This includes the activation of mTOR and pluripotency genes in high K⁺ conditions; mTOR genes show an enrichment in CRE sites in their promoters across all clusters (*p* < 0.0072, Odds Ratio 2.46, Fisher's Exact Test), pluripotency genes show enrichment for CRE sites in cluster A1 and A2 (*p* < 0.017, Odds Ratio 3.08, Fisher's Exact Test), and ETS sites in cluster A2 (*p* < 0.021, Odds Ratio 5.15, Fisher's Exact Test).

Turning to the genes activated later, particularly, those associated with pluripotency and germ-layer commitment in cluster A4, we find comprehensive enrichment of FOXH1, SOX and POU motifs in their promoters (Fig. 6d). These motifs correspond precisely with the early pluripotency TFs whose transcripts are activated in cluster A1 (Fig. 6e). Together, our high-resolution temporal profiling of the transcriptome in control and high K⁺ conditions reveals a cascade of transcriptional activation. We propose a model where a depolarized membrane opens VGCCs and elevates intracellular calcium leading to the expression of transcripts (including mTOR and pluripotency factors) whose promoters are enriched with calcium responsive motifs. This is followed by the activation of transcripts involved in pluripotency and germ-layer commitment, driven by the pluripotency factors activated in the early wave of gene expression.

Our transcriptome analysis not only revealed a potential gene regulatory network but pointed towards a role for mTOR. mTOR is critical for multiple cellular processes including autophagy, nutrient sensing, and an emerging role in pluripotency[50–56]. Because the expression of mTOR pathway members was increased in depolarizing conditions and pathways associated with mTOR, we reasoned that

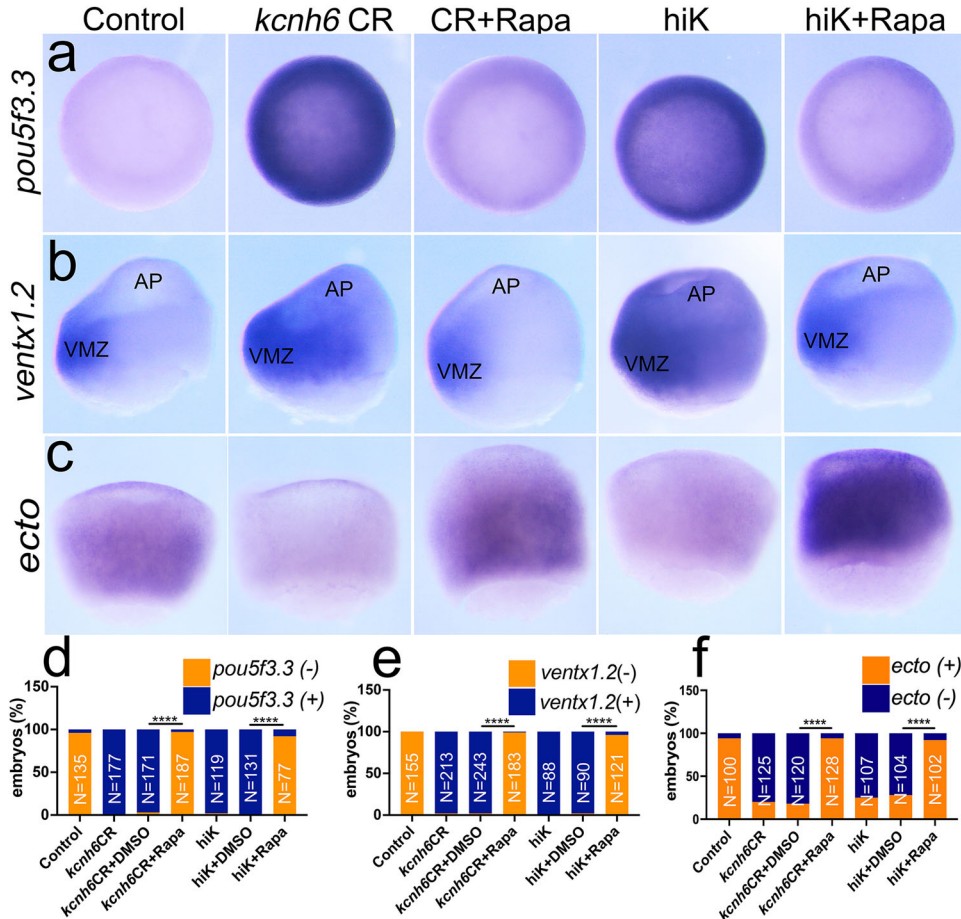

**Fig. 7 | V$_m$ polarization limits mTOR which promotes pluripotency. a**–**c** WMISH for pluripotency markers *pou5f3.3, ventx1.2,* and differentiation marker *ectodermin* in stage 10 embryos, depolarized by *kcnh6* depletion (*kcnh6*CR) or high K+ (hiK), and treated with vehicle (DMSO) or rapamycin (Rapa) as indicated; views are **a** animal pole; **b, c** lateral view with dorsal to the right; AP animal pole, VMZ ventral marginal zone. **d**–**f** Quantification of stage 10 embryos with present (+) or absent (−) expression of markers *pou5f3.3* (**d**)*, ventx1.2* (**e**), and *ectodemin* (**f**) in the animal pole area. *p*-values in **d** are: (*kcnh6*CR+Rapa vs *kcnh6*CR+DMSO) = 3.34e-087 and (hiK+Rapa vs hiK+DMSO) = 2.89e-047, in **e**: (*kcnh6*CR+Rapa vs *kcnh6*CR+DMSO) = 1.60e-110 and (hiK+Rapa vs hiK+DMSO) = 1.93e-050, and in **f**: (*kcnh6*CR+Rapa vs *kcnh6*CR+DMSO) = 3.38e-040 and (hiK+Rapa vs hiK+DMSO) = 4.89e-02; two-sided Fisher's exact test; total embryo numbers (N) in the graphs were collected over 3 independent experiments for *pou5f3.3* and *ventx1.2* and over 4 independent experiments for *ectodermin*; key for asterisks: ****$p \le 0.0001$. Source data are provided as a Source Data file.

mTOR signaling was upregulated and maintained pluripotency in these depolarized embryos. To test this hypothesis, we applied the mTORC1 inhibitor, rapamycin, to depolarized gastrulating embryos to see if this could abolish the aberrant expression of pluripotency markers *pou5f3.3* and *ventx1.2* in the animal pole and activate germ-layer differentiation. Rapamycin dramatically lowered expression of *pou5f3.3* and *ventx1.2* in the animal pole in *kcnh6* CR and high K$^+$ treated embryos compared to those embryos treated with vehicle alone and appeared comparable to untreated control embryos (Fig. 7a, b, d, e). Conversely, the expression of the ectodermal marker, *ectodermin*, which was reduced in depolarizing conditions (*kcnh6* depletion or exposure to high K$^+$), was recovered with rapamycin treatment (Fig. 7c, f). Therefore, a polarized V$_m$ at gastrulation onset is critical for limiting mTOR in order to suppress pluripotency genes and enter differentiation.

Finally, we tested whether our findings would also apply to human embryonic stem cells (hESCs). At stage 9, *Xenopus* animal cap cells are pluripotent in that they can, under appropriate conditions, form derivatives of any of the three germ layers (Fig. 4o). However, they contrast with hESCs in their limited capacity for self-renewal as the brisk pace of embryonic development proceeds. Therefore, we turned our attention to hESCs to test our findings in the context of a self-renewing pluripotent state and to determine their relevance to human

development. hESCs are already highly pluripotent, and we wondered if depolarization would lead to elevations in the pluripotency markers OCT4 and SOX2. Indeed, hESCs grown for two days with Ergtoxin to specifically block KCNH channels showed a modest but significant elevation of these markers over their already high levels in the pluripotency state as indicated by immunostaining and qPCR against these markers (Fig. 8a, b, Supplementary Fig. 8a, b). While not as specific as Ergtoxin for KCNH channels, Barium showed similar trends but did not rise to statistical significance. qPCR for the markers *OCT4*, *SOX2*, and *NANOG* also revealed upregulation of these genes at the transcript level, with *OCT4* and *SOX2* upregulated by day 2 and all three genes upregulated on day 5 (Supplementary Fig. 8a, b).

We also tested whether blocking K$^+$ channels with Ergtoxin affected the kinetics of differentiation. BMP4 induces differentiation to either mesodermal or extraembryonic fates in a dose-dependent manner[57–59]. Ergtoxin caused a significant delay in downregulation of pluripotency markers such as SOX2 at 12 h with a similar trend in NANOG (Supplementary Fig. 8c, d). As in *Xenopus*, the timing of differentiation of human embryonic stem cells appears significantly affected under depolarizing conditions.

To test whether the role of mTOR signaling downstream of membrane depolarization is conserved, we treated hESCs with rapamycin with or without Ergtoxin. Treatment with rapamycin led to

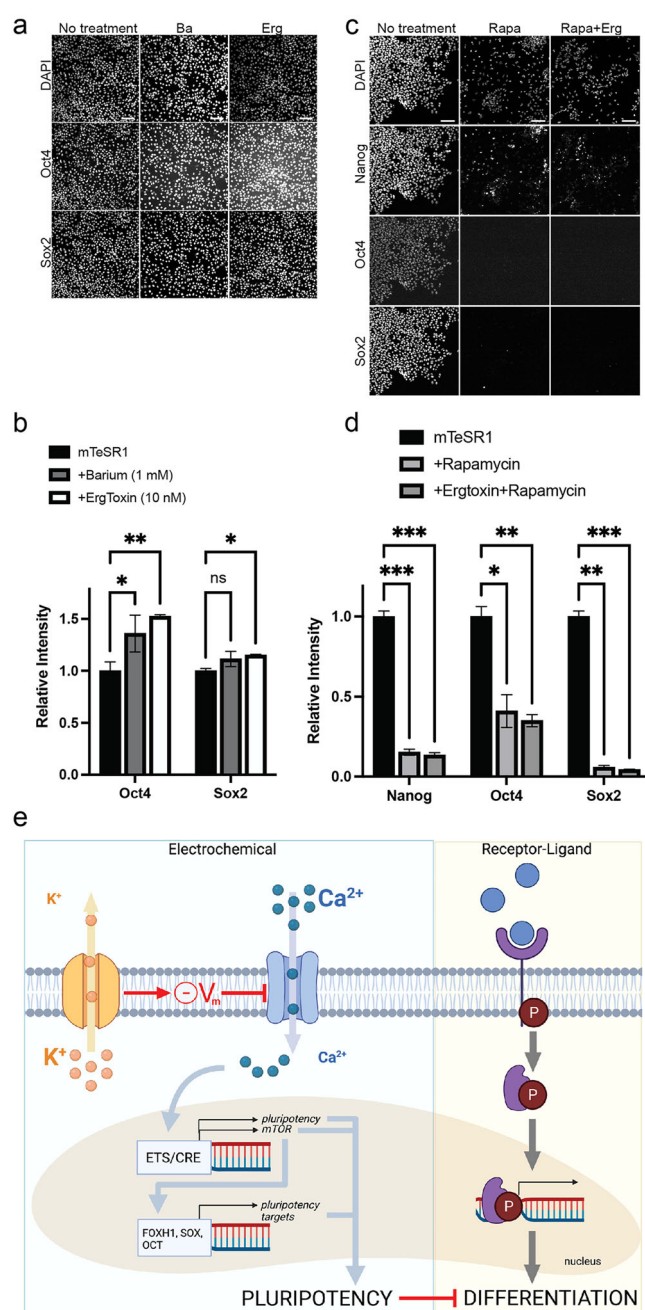

**Fig. 8 | Potassium channels affect pluripotency in hESCs. a** Images showing untreated hESCs grown in mTeSR1 media or cells treated with 1 mM Barium or 25 nM Ergtoxin and immunostained for pluripotency factors. Scale bar = 100 μm. **b** Quantification of the results in **a**; AU arbitrary units. *p*-values are Oct4: (control vs Ba) = 0.03, (control vs Erg) = 4.5e-04; Sox2 (control vs Ba) = 0.06, (control vs Erg) = 4.4e-04. **c** Images showing untreated hESCs grown in MEF-CM media or cells treated with 100 nM rapamycin with or without 25 nM Ergtoxin. Scale bar = 100 μm. **d** Quantification of the results in **c**; AU arbitrary units. *p*-values are Nanog: (control vs Rapa) = 4.5e-04, (control vs Rapa+Erg) = 4.3e-04; Oct4: (control vs Rapa) = 0.002, (control vs Rapa+Erg) = 0.002; Sox2 (control vs Rapa) = 7.7e-04, (control vs Rapa+Erg) = 7.6e-04. Graphs (**b**) and (**d**) present mean ± SEM; key for asterisks: *$p \leq 0.05$, **$p \leq 0.01$; ***$p \leq 0.001$; ns nonsignificant for $p < 0.05$; data were derived from 400 cells/3 independent replicates over at least 2 independent experiments. Source data are provided as a Source Data file. **e** Model for the onset of embryonic differentiation depicting classical biochemical signaling (right) that is complemented by regulation via membrane potential (left). In the electrophysiological pathway, potassium channels set the membrane potential, which limits activation of voltage-gated calcium channels and suppresses intracellular $Ca^{2+}$ levels. Both pathways result in changes in gene expression, mediated by intracellular signal transducers (right), or by factors that require calcium (left). While biochemical pathways are essential to induce expression of differentiation factors, the electrophysiological pathway affects cell fate indirectly by controlling the timing of downregulation of pluripotency genes. Adapted from "Transporters", by BioRender.com (2022). Retrieved from https://app.biorender.com/biorender-templates.

activate a program of cellular differentiation (Fig. 8e). The $K^+$ and $Ca^{2+}$ channel network upstream of pluripotency factor expression potentially represents an extremely robust control mechanism over the first stages of organism development: ion channels are effective at low expression levels, modular and thus partially redundant with respect to each other (i.e., subunits of different channels can heterodimerize to form a channel if subunits of the same channel are unavailable), and their function depends on the existence of a simple ion concentration gradient across the plasma membrane. The basic elements of this regulation, $K^+$ and $Ca^{2+}$, are readily available extracellularly, conferring this system with some independence from protein-dependent cell signaling. Ultimately, ion channel networks may represent an additional mechanism to regulate cell fate during development, which is complementary to the established paradigm of gene expression regulation by secreted factors and ligand-receptor signaling (Fig. 8e).

Our work and the work by others supports the notion that membrane potential modulates development in a variety of contexts including LR patterning[2–6], chondrocyte differentiation in chick limbs[13] as well as morphogenesis of the developing Drosophila wing[8,9]. Additionally, our work adds to the molecular mechanisms that have associated membrane potential with stem cells differentiation[12,14,60–62]. Of note, previous work has suggested that $K^+$ itself may have signaling properties that affect the directed differentiation of hESCs[63]. However, our study indicates that it is $V_m$ and not $K^+$ itself that is critical. This is consistent with the notion that, because $K^+$ is so abundant inside the cell, it is unlikely to act as a signaling moiety, unlike $Ca^{2+}$.

While elevations in intracellular calcium do have physiological functions in the prospective mesoderm and ectoderm of the gastrula, both in patterning[24,25] and morphogenesis[30], we show that regulation of voltage-gated calcium channels by $V_m$ is critical for the exit from pluripotency. Based on this, we suggest that low intracellular calcium reduces the expression of mTOR and pluripotency factors, which is conducive to differentiation onset. While we have not eliminated other voltage responsive signaling, our results indicate that calcium is critical in this context.

Regulation by $V_m$ and $Ca^{2+}$ presents interesting implications for developmental timing, as they appear to be key in specifying the exact time frame of pluripotency exit. Indeed, previous work has also implicated mTOR in developmental timing of the pluripotent state[64],

reduction of pluripotency markers in a dose-dependent manner with near complete loss by 5 days (Fig. 8c). In the presence of rapamycin, Ergtoxin had no effect on pluripotency markers (Fig. 8d), indicating that mTOR is downstream of membrane depolarization in hESCs as in *Xenopus*. Although rapamycin did reduce the final cell number per well, it inhibited the effect of Ergtoxin on pluripotency marker expression independently of the density at which cells were seeded (Supplementary Fig. 8e) and of the final cell number in the well (Supplementary Fig. 8f). Taken together, our data support that the polarization of membrane potential via KCNH channels promotes the exit from pluripotency and the activation of differentiated cell fates in *Xenopus* and human cells.

## Discussion

We propose a model in which membrane voltage regulates intracellular calcium during a critical stage of embryonic development, at which point cells need to extinguish pluripotency factors in order to

and our work demonstrates the upstream regulation of mTOR in this context. The requirement of $V_m$ for this developmental period is also underscored by measurements in developing *Xenopus*[16] and *Triturus*[15] showing a clear hyperpolarization of $V_m$ towards the onset of germ-layer differentiation that is concurrent with a reduction in calcium levels[26]. Therefore, it may be that membrane polarization may act as a timing mechanism to allow embryonic cells to be synchronized before the onset of differentiation. To do so, the embryo employs ion channels, mTOR, and an array of pluripotency transcription factors. Then secreted factors can act on these pluripotent cells to activate different programs of differentiation across the embryo (Fig. 8e). Of note, tissues with abnormal ion homeostasis at gastrulation onset do have differentiation potential and may eventually differentiate at later stages, but the significant delay in exiting pluripotency dramatically affects gastrulation morphogenesis itself as well as LR patterning.

Importantly, genetic data from Htx/CHD patients identified the KCNH family and especially *KCNH6*. Recent work has identified ion channels as an interesting intersection between CHD and autism[65]. While ion channels in neuronal development are well established[14], our work adds to the role of ion channels in early embryonic development that includes pluripotency, LR patterning, and congenital heart disease.

Finally, the induction and maintenance of pluripotency clearly depends on a set of transcription factors, and our work connects $V_m$ and intracellular calcium as upstream of this pluripotency program. This finding may be applicable to multiple contexts in which stem cells play a role in embryonic development or adult tissue homeostasis. Our work demonstrates the importance of $V_m$ in vivo during early embryonic development as well as in vitro in human stem cells. Importantly, this pathway is readily manipulated by a wide range of highly specific channel inhibitors or simple changes in extracellular ionic concentrations. Therefore, we define multiple tools for pluripotency manipulations in embryos, organoids, and adult tissues where stem cells play a critical role.

## Methods

### *Xenopus* husbandry

In this study, we used *Xenopus tropicalis* from the N (Nigerian) strain. Adult *X. tropicalis* were raised and housed according to our established protocols which were approved by the Yale Institutional Animal Care and Use Committee. We induced ovulation, performed IVF, and raised embryos in 1/9× MR as previously described[66]. We staged *X. tropicalis* embryos according to Nieuwkoop and Faber[67]. In *Xenopus*, sex is determined in a 50:50 ratio genetically, and our experiments were all performed in embryos under 72 h of age before the sex of the animal can be determined morphologically.

### Morpholino oligonucleotides, mRNA, and CRISPRs

All injections of *Xenopus* embryos were performed at the one-cell stage using a fine glass needle and Picospritzer system, as previously described[68]. A *kcnh6* translation blocking (*kcnh6* MO, 5′-GGTCCTCGAAGTTTAGGATAAACAT-3′) and a scrambled morpholino oligonucleotide were obtained from Gene Tools LLC and injected at 10 ng to deplete *kcnh6* or as a control, respectively. CRISPR sgRNAs for *kcnh6* targeted either exon 3 or exon 4 based on the v7.1 gene model of the *X. tropicalis* genome (CRex3: 5′-GGAATAAGGGGTGAAGACAGCGG-3′ and CRex4: 5′-AGGGCGCTCTACATTTCCAATGG-3′). CRISPR sgRNAs for *cacna1c* (5′-GCAGACGGGGGCAGCGCCATTGG-3′) and *cacna1g* (5′-GGTTAATGGCTCTCAGCGGGCGG-3′) were designed from the v7.1 model of the *Xenopus tropicalis* genome. For F0 CRISPR knockdown, embryos were injected with 1.5 ng Cas9 Protein (PNA-Bio) and 400 pg of targeting sgRNA and raised to desired stages as previously described[69]. For *pitx2* and *coco* expression analyses, the dose of *kcnh6* sgRNA was halved to a subphenotypic dose of 200 pg to obtain embryos without gross morphological gastrulation defects. Full-length

human *KCNH6* (NM_030779.3 cloned in pCS107), *GCaMP6* (subcloned in pCSDest) and *mCherry* cDNAs (Addgene #34935; in pCS2 +), were used to generate capped mRNAs in vitro by first linearizing with appropriate restriction enzymes and then transcribing with the mMessage machine kit (Ambion). mRNAs were injected at 3 pg (human *KCNH6*), 150 pg (*GCaMP6*), and 150 pg (*mCherry*) per embryo. Embryos were raised at 21 °C to allow time for sufficient expression levels at blastula/gastrula stages.

### Inference of CRISPR edits (ICE) analysis

Genomic DNA from CRISPR and control embryos were obtained by lysing individual, stage 45 tadpoles in 50 mM NaOH as previously described[69] and amplifying PCR fragments around the CRISPR target site that encompass ~200 bp upstream and 500 bp downstream of the site. The following primers were used for CRISPRs targeting exons 3 and 4 of the *kcnh6* locus, respectively: CRex3-F: 5′-CAGGACTGATGAAAGCAAGC-3′ and CRex3-R: 5′-GCTTATCCATAGCTGTAACAACG-3′; CRex4-F: 5′-GAGACAGTAGGCTGTTCC-3′ and CRex4-R: 5′-CCACAAGCAGTTTCACTACC-3′. PCR fragments were Sanger sequenced using the same forward primers, and sequencing traces were uploaded for analysis with the Synthego ICE analysis web tool to assess editing outcomes.

### Organ *situs*

Stage 45 *Xenopus* embryos were paralyzed with benzocaine or tricaine and scored with a light stereomicroscope. Cardiac looping was determined by position of the outflow tract; D-loop: rightward, L-loop: leftward; A-loop: midline. Normal intestinal looping was scored as counter-clockwise rotation of the gut, while abnormal intestinal looping was scored as completely inverse gut rotation (clockwise) or complete lack of looping (unlooped). While a completely inverted gut rotation is clearly an abnormality of LR patterning, an unlooped gut is less clear so we only considered an unlooped gut as abnormal *situs* when combined with abnormal placement (left-sided or midline) of the gall bladder. To quantify total abnormal organ *situs*, each tadpole was counted only once, regardless of whether multiple organs were affected.

### Whole mount in situ hybridization

Digoxigenin-labeled antisense probes for *pitx2* (TNeu083k20), *dand5/coco* (TEgg007d24), *myoD* (Tneu017H11), *myf5* (TGas127b01), *tbxt* (TNeu024F07), *foxj1* (Tneu058M03), *ectodermin* (TNeu104j16), *foxI1a* (Tgas002H16), *mixer* (TGas105b05), *vegT* (TGas066f22), *gsc* (TNeu077f20), *xnr3* (Tgas011k18), *vent2* (BG885317), *oct25* (TGas051h05), *oct60* (IMAGE: 7526158), *oct91* (IMAGE: 7575764), *vent1* (BG487195), *sox2* (Tgas061h22), *cytokeratin* (IMAGE:6991625), and *sox17β* (BG886038) were in vitro transcribed using T7 High Yield RNA Synthesis Kit (E2040S) from New England Biolabs. In order to generate a full-length antisense probe for *X. tropicalis kcnh6*, *kcnh6* cDNA was cloned from stage 45 tadpole whole mRNA using primers xtkcnh6-F: 5′- ATGTTTATCCTAAACTTCGAGGACC-3′ and xtkcnh6-R: 5′-CTAACTTCCTGGAAGACCTGGG-3′ (XM_012952904.1). We note that *kcnh6* had been misannotated as *kcnh2* in the v7.1 model of the *X. tropicalis* genome (We used NCBI Annotation XP_012808358.2 to identify KCNH6). Embryos were collected at the desired stages, fixed in MEMFA for 1–2 h at room temperature (RT) and dehydrated in 100% ethanol. GRPs were dissected post fixation and prior to dehydration to detect *dand5*. To detect putative gene expression in the prospective endoderm (*mixer*, *vegT*, *kcnh6*) gastrula stage embryos were bisected to facilitate better probe access. Briefly, whole mount in situ hybridization of digoxigenin-labeled antisense probes was performed overnight, the labeled embryos were then washed, incubated with anti-digoxigenin-AP Fab fragments (Roche 11093274910), and signal was detected using BM-purple (Roche 11442074001), as previously described in detail[68].

## Medium conditions and treatments of embryos

Normal embryonic medium is 1/9x modified Ringer's (MR) containing 11 mM NaCl, 0.2 mM KCl, 0.2 mM CaCl$_2$, 0.1 mM Mg$_2$Cl and 0.55 mM HEPES. To allow Ergtoxin to penetrate the embryos, we manually removed the vitelline envelope of stage 8 embryos and incubated embryos in 1/9× MR containing 50 nM Ergtoxin (Alomone STE-450) until stage 12. Embryos were then transferred back into 1/9× MR lacking Ergtoxin to develop until stage 45 in order to score organ *situs*. Barium chloride was applied into the medium at 20 mM and embryos were thoroughly rinsed in 1/9× MR after each incubation period for further development in Ba$^{2+}$-free medium. For extracellular K$^+$ manipulations, the KCl concentration in 1/9x MR was modified from 0.2 mM (normal) to 20 mM (high) or 0 mM (low). The ionophore valinomycin (ACROS) and L-type VGCC blocker Nifedipine (ACROS) were diluted in DMSO as stock solutions and applied to embryos in 1/9× MR at 2 and 10 µM, respectively. Treatments performed during gastrulation were applied from stage 8 through stage 12, and embryos were then rinsed thoroughly and returned into 1/9× MR. For rapamycin, we created a standard stock solution of 50 mg/ml in DMSO. The stock solution was diluted 1:2500 in the appropriate embryonic media (final 20 µg/ml). Embryos were treated at stage 7 and then fixed at stage 11 for in situ hybridization.

## GRP immunofluorescence

Embryos were fixed at stage 17 in 4% paraformaldehyde-PBS for 2 h at RT, washed in PBS, and then dissected to obtain GRPs. GRPs were permeabilized for 30 min at RT using 0.1% Triton-PBS (PBST), then blocked in 1% BSA-PBST for 1 h at RT and incubated in primary antibodies diluted in 1% BSA-PBST overnight at 4 °C (anti-myoD 1:100; LsBio C143580-100 or anti-acetylated tubulin 1:2000 Sigma T-6793). GRPs were then washed in PBST for 30 min and then incubated with secondary antibodies in 1% BSA-PBST for 1 h at RT. Phalloidin (1:50; Molecular Probes) and Hoechst 33342 (1:1000; Molecular Probes) were diluted into the secondary antibody solution. Images were acquired using a ZEISS 710 laser scanning confocal microscope.

## Intracellular V$_m$ recordings

For recordings, devitellinized, stage 10 *kcnh6* or control MO-injected embryos were mounted into non-toxic clay with their animal pole exposed and covered with 1/9x MR. To investigate the resting potential, animal pole cells were impaled with a high-impedance (-70 MΩ), sharp microelectrode filled with 3 M KCl for intracellular recordings. The recordings were made using an Axon 200B amplifier and digitized using a Digidata 1320 digitizer. Jclamp software for Windows was used in current clamp mode. All electrodes were zeroed just before entry into the cells.

For the series of intracellular recordings in high K$^+$ and choline treated embryos, stage 8–9 embryos were impaled similarly with an electrode of -40 MΩ. These recordings were made using a HEKA EPC10 amplifier. We used HEKA PatchMaster v2x67software for Windows. All electrodes were zeroed just before entry into the cells.

## Calcium imaging

GCaMP6 and mCherry mRNAs were mixed and injected into embryos at the one-cell stage. Half of these embryos were then injected with *kcnh6* MO and the other half with control MO, still at the one-cell stage. Embryos were transferred at stage 10 into the round wells of a press-to-seal silicone isolator (Sigma S3685) mounted between two cover slips in 2% Methylcellulose-1/9x MR. GcaMP6 and mCherry fluorescence was then captured for 20 s (1 frame per second) via time lapse in the whole animal pole of each embryo with a ×20 objective of an LSM710 confocal microscope using identical acquisition settings across Control MO and *kcnh6* MO embryos. Time lapse recordings were conducted randomly and in an unbiased manner in regard to presence and intensity of

calcium transients. However, all embryos did display transient increases in GcaMP6 fluorescence, varying in intensity and spreading to multiple cells. The frames of each recording were sorted to identify the calcium transient peak (in area), and GcaMP6 fluorescence intensity was quantified at peak as a ratio to mCherry in mCherry+ cells. The maximum Ca$^{2+}$ transient area was calculated by demarcating in Fuji the GcaMP6(+) vs GcaMP6(−) area of the animal pole at transient peak. To avoid mosaicism artifacts, only embryos with even, non-mosaic mCherry expression across the entire animal pole were considered. To avoid embryonic stage dependent fluctuations in Ca$^{2+}$ transient size, we verified each embryo for stage by progression of blastopore closure and alternated recordings of control and *kcnh6* MO embryos. Of note, there were no notable differences in mCherry expression between Control MO and *kcnh6* MO embryos.

## Animal cap pluripotency assays

After manually removing the vitelline envelope of stage 9 or 12 embryos, animal caps were excised and placed on agarose coated dishes in 1/9× MR solution. Caps were then directly placed into agarose coated wells of a 96-well plate in 1/3× MR containing 0.1% BSA and cultured without activin to allow for differentiation into epidermis, with low (20 ng/ml) activin to induce mesoderm, or high activin (200 ng/ml) to induce endoderm, as previously described[43]. Explants were raised at 25 °C until reaching the equivalent of stage 18 (monitored in whole embryos of the same batch), then fixed in 4% paraformaldehyde, washed in PBS, bleached to eliminate pigmentation (0.5× SSC, 5% formamide, 1.2% H$_2$O$_2$), and then processed by in situ hybridization as described above.

## RNA-seq

For RNA-Seq, embryos were kept at 25 °C either in 1/9× MR or in 10 mM KCl solution, and 10 embryos were harvested per time point and condition every 30 min starting at stage 8 and concluding at stage 13. Samples were immediately frozen and kept at −80 °C until homogenized in 100 µl Trizol spiked with ERCC RNA Spike-In Mix. 10 µl ERCC RNA Spike-In Mix (Thermo Fisher Scientific) were first diluted into a final volume of 870 µl DEPC water and then further diluted 1:10 into Trizol, which was used to homogenize the samples. Total RNA was purified from the embryo Trizol homogenates according to the manufacturer's recommendations. After isopropanol precipitation, RNAs were resuspended in DEPC water and any contaminating genomic DNA was removed by overnight precipitation in 5 M LiCl at 4 °C. RNA was subsequently pelleted and washed twice with 70% ethanol. All RNAs were resuspended in DEPC water (2 µl/embryo), and finally, RNA quality was verified by Bioanalyzer. All libraries were sequenced with 100 bp paired-ends on an Illumina NovaSeq6000.

## Quantification RNA-seq

Stranded paired-end 100 bp RNA-seq reads were aligned to the Xt9.1 genome combined with ERCC spikes using STAR[70] and quantified as transcripts per million (TPM) for each isoform with RSEM[71] using the RSEM-STAR pipeline, with additional options "--seed 1618 --calc-pme --calc-ci --estimate-rspd --paired-end". Using the ERCC spikes we identified a batch-dependent GC bias where AT-rich transcripts were preferentially lost as compared to GC-rich transcripts (Supplementary Fig. 7b). We leveraged knowledge of spike-in concentrations to build a GC model correction based on the dinucleotide content of RNAs. We calculate the propensity of each of the 16 dinucleotides (*AA, AC, ..., TG, TT*) within each spike sequence, with $f_{ik}$ is the frequency of dinucleotide $k$ within sequence $i$, its propensity is $p_{ik} = f_{ik}/\sum_j f_{ij}$. We then employ the following linear model to correct the TPM $t_{si}$ of RNA spike $i$ in sample $s$ to its known concentration $c_i$:

$$\log c_i = \alpha + \beta_T \log t_{si} + \sum_j \beta_j \log \rho_{ij}$$

We use the GLM.jl (https://github.com/JuliaStats/GLM.jl) in the Julia language to apply this model and add a pseudocount of 2 to all dinucleotide frequencies. As the GC effect varies between UC and high K+ samples (Supplementary Fig. 7b), we apply the correction independently to UC and High K+. The correction is able to explain a significant proportion of variance in spike TPM, increasing $R^2$ from 0.807 and 0.733 to 0.965 and 0.964 respectively from UC and high K+ samples. We apply this correction to each isoform of all genes quantified with the dinucleotide propensities of each isoform and RSEM isoform quantifications. We then sum all corrected quantifications at the isoform level to derive gene level quantifications. This allows us to account for differing isoforms of the same gene with differing dinucleotide propensities.

### Filtering of genes for differential expression analysis

We first filtered 34,192 quantified genes to find those with sufficient temporal expression for further analysis, we selected genes that had runs of 6 consecutive samples with uncorrected TPM > 0.4. This resulted 13,310 from which we excluded a further 162 genes which where excessively altered by the above described correction procedure, these had log2 fold changes between corrected or uncorrected quantifications outside of the interval (−2.5, 4.5). After dinucleotide correction and filtering we found excellent concordance between samples, with minimal evidence of outlying samples, by Spearman Correlation comparisons and principal components analysis (PCA) (Supplementary Fig. 7c, d). The two domains in visible in pairwise Spearman comparisons (Supplementary Fig. 7c) reflect the loss of maternal RNA and the commencement of widespread zygotic transcription as we previously described[19]. Projection onto the first two principal components revealed that samples lie in appropriate order on a trajectory in 2D space, and the largest divergences between UC and high K+ occur midway through the time series in agreement with Gaussian process differential expression and clustering described below. Corrected dinucleotide gene expression abundances are used in all analyses.

### Temporal differentiation expression

To determine genes temporally differentially expressed we used Gaussian process (GP) regression as we have previously applied[19]. All GP regression was performed with GaussianProcesses.jl (https://github.com/STOR-i/GaussianProcesses.jl; https://arxiv.org/abs/1812.09064). Due to the overdispersed nature of RNA-seq count data, we apply a variance stabilizing transform that puts all genes on the same scale: $y_{si} = \sqrt{\alpha + \beta x_{si}/m_i}$, with $x_{si}$ the dinucleotide corrected abundance of gene $i$ in sample $s$, $m_i$ the maximum $x_{si}$ over all samples, and $\alpha = 1$, $\beta = 1000$. We then perform exact GP regression (GP prior and a Gaussian likelihood) with Matern52 kernel, we optimize the three associated hyperparameters: $\sigma_f^2$ the signal variance, $\tau$ the timescale (using previous terminology[19], this parameter is commonly referred to as the lengthscale $l$), and $\sigma_n^2$ the sample noise variance. Parameters are selected by optimizing marginal log-likelihood with parameters in log space: $\log\sigma_f, \log\tau, \log\sigma_n$, and to ensure physiologically reasonable values for each we place Gaussian priors, $\mathcal{N}(\mu,\sigma)$ over each of these variables respectively $\mathcal{N}(1.4,4.0), \mathcal{N}(1.2,1.0), \mathcal{N}(1.0,0.75)$. Finally, we reported GP median and 95% confidence intervals through our inverted data transformation $\hat{x}_{si} = m_i(\hat{y}_{si}^2 - \alpha)/\beta$ and set $\hat{x}_{si} = 0$ for $\hat{y}_{si} < \sqrt{\alpha}$.

To determine temporal differential expression, we calculate a marginal likelihood ratio for whether we prefer separate GP models for UC and high K+ or a single GP model for all data combined. If $L_U$ and $L_K$ are the marginal log-likelihoods for UC and high K+ respectively, and $L_{UK}$ is the marginal log-likelihood for a single regression through UC and high K+ together. Then we calculate log-likelihood ratio $LR = L_U + L_K - L_{UK}$ of evidence in favor of two models (essentially that the UC and High K+ have different expression trajectories for a given gene) and determine genes with $LR > 0$ as temporally differentially

expressed. This resulted in 5144 differentially expressed genes, with 4043 activated and 1101 repressed (Supplementary Fig. 7d which shows that the max absolute divergence z-score between UC and high K+ trajectories increases with $LR$). We also considered a more stringent condition for differential expression using the Bayesian Information Criterion (BIC)[19], that resulted in 2388 differentially genes. We found that this diminished differential expression gene set enrichments described below, indicating that the BIC was too conservative and we continued with our condition based on log-likelihood ratio.

### Clustering

To determine sets of differentially expressed genes with similar trajectories, we applied K-means clustering to activated and repressed genes independently. We define a gene as activated if Gaussian process median for High K+ exceeds UC on average, and repressed if it does not, we found no genes for which the mean of High+ and UC differences was zero. We cluster UC and High K+ genes by taking Gaussian process medians and normalizing by the maximum value experience by UC or High K+. We then cluster both trajectories together employing the kmeans function offered by Clustering.jl (https://github.com/JuliaStats/Clustering.jl) with default settings and random seed 16. To select the cluster number, we calculated the silhouette score for activated and repressed clusters for k = 2–10. We found that the maximal mean silhouette score activated genes was k = 3 and for repressed genes was k = 2, but that scores were broadly similar for k = 2–4 and decreased significantly for k > 4, suggesting that k = 4 provides a reasonable partition of the data. In line with this we explored the clusters from k = 2–10, and found that key clusters were not well-resolved for k < 4 and that k > 4 clusters refined k = 4 behaviors. As k > 4 did not reveal new behaviors and did not improve gene set enrichments, we selected k = 4 to cluster activated and repressed genes.

### Gene set enrichments

To assess the composition of each cluster we performed gene set enrichments using Enrichr[45]. We took genes from each cluster with a known *Xenopus* gene symbol and converted these to human symbols, by removing any ".N" suffix for an integer N (for example, ventx1.1 becomes ventx1) and converting to uppercase. We then made the following substitutions to convert certain known *Xenopus* gene symbols to human where the name of the ortholog has diverged or only a paralog exists: pou5f3 → POU5F1, mix1 → mixl1, dppa2 → DPPA4, lefty → lefty2, ventx1-3 → NANOG, mespb → MESP1, sox17a/b → SOX17. We remove any duplicate names that arose in this process. We calculated enrichments for the following gene sets: KEGG_2019_Human, BioPlanet_2019, WikiPathways_2019_Human, GO_Biological_Process_2018, GO_Molecular_Function_2018, GO_Cellular_Component_2018, ChEA_2016. We calculate enrichments for each cluster individually and consecutive combinations of the 4 clusters: 1, 2, 3, 4, 12, 23, 34, 123, 234, 1234. All enrichments can be found in Supplementary Data 1, selected enrichments are given in Fig. 6C and terms are given shortened labels for brevity (Table 2):

### Motif analysis

To find motifs enriched in the promoters, we took the 500 bp upstream of the promoter of the maximally expressed isoform for each gene in the four activated and four repressed clusters, along with a background of the 500 bp upstream of all annotated TSS in the Xt9.1 genome. We extracted fasta files for each of these sets of regions, and then used findMotifs.pl from Homer[72] to search for known motifs with options: "findMotifs.pl cluster*AB*.fa fasta out*AB* −fasta background.fa −nomotif" where $A \in \{\text{activated, repressed}\}$ and $B \in \{1, 2, 3, 4\}$. We filtered results to select best matching motifs from related families, namely we collapsed all ETS motifs to the canonical Homer ETS promoter motif; all SP and KLF motifs to SP1; SOX motifs to SOX2; all HOX

**Table 2 | Selected enrichments from RNA-seq time course**

| Label | Term | Gene set |
|---|---|---|
| mTOR | mTOR signaling pathway | KEGG_2019_Human |
| Autophagy | Autophagy | KEGG_2019_Human |
| Ubiquitin | Ubiquitin-protein transfer-ase activity GO:0004842 | GO_Molecular_Function_2018 |
| ER | Protein processing in endo-plasmic reticulum | KEGG_2019_Human |
| Spliceosome | Spliceosome | KEGG_2019_Human |
| WNT signaling | Wnt signaling pathway | KEGG_2019_Human |
| Pluripotency signaling | Signaling pathways reg-ulating pluripotency of stem cells | KEGG_2019_Human |
| Endoderm differentiation | Endoderm differentiation WP2853 | WikiPathways_2019_Human |

Selected enrichments and their associated abbreviated label listed in Fig. 6c.

motifs to HOXD13 (the highest scoring HOX); we represent all GFY and Ronin matches as ZNF143 (for which the motifs overlap); and we excluded motif annotated as PRDM10, due to low confidence in the motif. The motif annotated as ATF1 is an example of the cAMP response element (CRE) bound by CREB factors including ATF1, we label this as CRE/ATF1. The top 16 motif enrichments are given in Supplementary Fig. 7h, in Fig. 6d we give the top 6 motif matches, excluding ZNF143 due to divergence from the JASPAR database, to which we add CRE/ATF1 as a putative calcium responsive element motivated by CREM/CREB1 gene set enrichments Supplementary Fig. 7j).

To calculate CRE and ETS motif enrichment for mTOR and pluripotency genes, we took genes annotated with the terms *mTOR* signaling pathway and Signaling pathways regulating pluripotency of stem cells from KEGG_2019_Human as provided by Enrichr[45] that are activated in high K+ (LR > 0) and are present in clusters A1 and A2. The resulting genes were subjected to the same promoter analysis, using Homer to calculate the occurrence of the maximal ATF/CRE family motif and the ETS motif in these promoters and the background set to report Fisher Exact test p-values and Odds Ratios.

### *Xenopus* biological replicates, statistical methods, graphs, and models

In experiments where embryos were evaluated for phenotypes and scored (gastrulation, left-right patterning, in situ hybridizations) we carried out three to five biological replicates and Fisher's exact test to evaluate statistical significance. The animal cap experiment was performed twice with a total score of four to eight animal caps per experiment. For the calcium transient analyses, data was collected from three to five embryos in each experiment in three independent experiments, and statistical analyses on GCaMP/mCherry fluorescence intensity as well as $Ca^{2+}$ transient area were performed using student's *t*-test. For whole cell electrophysiological recordings, three to five embryos (two cells each) were examined for their membrane potential and statistical significance was tested by student's *t*-test. Graphs were designed using GraphPad Prism software. Models were created with BioRender.com.

### hESC culture

hESCs were grown in mTeSR1 (STEMCELL Technologies) in tissue culture dishes coated with Matrigel (Corning; 1:200 in DMEM/F12) and kept at 37 °C, 5% $CO_2$. The cell lines used were ESI017 (ESIBIO) and H9. Cells were routinely passaged using dispase (STEMCELL Technologies) and tested for mycoplasma contamination and found negative. For rapamycin experiments, cells were grown in MEF-conditioned HUESM media supplemented with 20 ng/ml bFGF as previously described[73]

with or without 100 nM rapamycin, which we found to increase the survival of rapamycin treated cells compared to cells grown in mTeSR1.

### hESC treatments and differentiation

Cells were dissociated with accutase and seeded onto eight-well imaging slides (ibidi 80826) at a density of $4-6 \times 10^4/cm^2$. Cells were seeded and maintained in Rock-inhibitor Y27672 (MCE; 10 µM) to increase survival and the uniformity of response. Treatments with 1 mM $BaCl_2$ or 10 or 25 nM Ergtoxin or 100 nM rapamycin were initiated 4 h after seeding. Differentiation was initiated 24 h after seeding where indicated. To initiate differentiation, the media was replenished with/ without $BaCl_2$ or Ergtoxin and treated with the indicated growth factors or small molecules. The media with any treatments was replenished daily.

### Immunofluorescence of hESCs

Cells were fixed for 30 min in 4% paraformaldehyde, rinsed twice with DPBS (without $Ca^{2+}$ and $Mg^{2+}$, denoted DPBS-/-), and blocked for 30 min at room temperature. The blocking solution contained 3% donkey serum and 0.1% Triton X-100 in 1× $DPBS^{-/-}$. After blocking, the cells were incubated with primary antibodies at room temperature for 2 hours. Antibodies and concentrations are listed in Supplementary Table 1. Cells were washed three times with DPBST (1X DPBS-/- with 0.1% Tween 20) and incubated with secondary antibodies (AlexaFluor 488 A21206, AlexaFluor 555 A31570 and A21432, and AlexaFluor 647 A31571, Thermo Fisher; 1:500) and DAPI for 30 min at room temperature. After secondary antibody incubation, samples were washed in DPBST and then DPBS at room temperature.

### hESC Imaging and analysis

Images were acquired using a ×20, NA 0.75 objective on an Olympus IX83 inverted epifluorescence microscope or an Olympus/Andor spinning disk confocal microscope. Cell segmentation was performed using ilastik software[74]. This segmentation was cleaned (to remove debris and to separate merged cells) and mean nuclear protein intensities as well as standard errors were quantified using a custom MATLAB code. Nuclear intensities were normalized by DAPI to correct for intensity variation due to optics. Code is available at https://github.com/warmflashlab/Sempou2022_Code.

### qPCR

For qPCR, hESCs were grown with or without ErgToxin (25 nM) for the indicated times. RNA collection and DNase treatment were performed using the RNAqueous®-Micro Total RNA Isolation Kit (AM1931) and cDNA was synthesized with the SuperScript Vilo cDNA Synthesis Kit (Fisher Scientific 11754-050). qPCR measurements were collected using SYBR Green reagent (LifeTech-4367659) on a Step OnePlus instrument (Applied Biosciences). Data were normalized using the housekeeping gene GAPDH. Primers for qPCR were:

*OCT4*: 5'-caagctcctgaagcagaagag–3', 5'-ccaaacgaccatctgccgcttt–3',
*SOX2*: 5'-ccatgcaggttgacaccgttg–3', 5'-tcggcagactgattcaaataata-3',
*NANOG*: 5'-tgggatttacaggcctgagcca–3', 5'-aagcaaagcctcccaatcc-caa–3',
GAPDH: 5'-caccgtcaaggctgagaacg-3', 5'-gccccacttgattttggagg-3'.

### Statistics and reproducibility

Embryo and cell sample sizes in this study were chosen according to the standards in the field. No data were excluded from the analyses. The investigators were blinded to allocation during experiments and outcome assessment whenever feasible.

For statistics, two-sided Fisher's exact test was used to assess significance of two variables with independent proportions, (treated/untreated) and their outcomes (normal/abnormal). This test was applied when scoring for morphological phenotypes of embryos and

WMISH results, under control conditions or after microinjection (CR, MO, mRNA), medium manipulations or treatment with compounds. Two-tailed, unpaired $t$-tests or ANOVA were used to test whether the means of two populations differ largely from one another and were employed for $V_m$, calcium intensity and calcium transient area comparisons as well as all hESC experiments.

## Reporting summary

Further information on research design is available in the Nature Research Reporting Summary linked to this article.

## Data availability

The RNA-seq time series of uninjected-control (UC) and high K+ embryos generated in this study have been deposited in the Gene Expression Omnibus under accession GSE186670. Source data are provided with this paper.

## Code availability

All source data and code required to analyze RNA-seq time series and generate figures are available at https://github.com/owensnick/KCNH6GenomicsFigures.jl, https://doi.org/10.5281/zenodo.6967350. Source code for quantitating protein intensities in human ES cells is available at https://github.com/warmflashlab/Sempou2022_Code.

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

## Acknowledgements

We would like to thank the patients and their families, who are the inspiration for this study. We would like to thank Michael Slocum, Maura Lane, and Sarah Kubek for frog care and husbandry, Richard Harland (especially Debbie Pangilinan) and Addgene for providing plasmids, and Sergei Sokol for kindly supplying activin. We thank the Yale Center for Cellular Imaging for microscopy support, and Yale Center for Genome Analysis for RNA-seq. E.S. was supported by a Brown-Coxe Postdoctoral Fellowship. N.O. is supported by an Expanding Excellence in England Lectureship. This work was supported by the NIH R01HD081379 to M.K.K., NIH R01DK072612 to M.J.C., NIH R01EY021195 to D.Z., and NIH R01GM126122, NSF MCB-1553228, and Welch Foundation C-2021 to A.W. M.K.K. was a Mallinckrodt Scholar.

## Author contributions

E.S., M.J.C., D.Z., A.W., N.D.L.O., and M.K.K. conceived and designed the experiments and analyzed data from the experiments. E.S., V.K., W.H., and M.K.K. performed all *Xenopus* experiments. E.S., J.Z., L.T., and D.Z.

performed electrophysiological recordings in *Xenopus*. N.D.L.O. analyzed all genomic data. E.S., C.G., A.W., and M.K.K. conceived, designed, and analyzed experiments in human embryonic stem cells. C.G., E.C.-A., and A.W. performed human embryonic stem cell experiments. The manuscript was written by E.S., A.W., N.D.L.O., and M.K.K. and all authors critically evaluated the manuscript.

## Competing interests

The authors declare the following competing interests: M.K.K. is a Founder and President of Victory Genomics, Inc. Yale University has filed a provisional patent application related to this work. The remaining authors declare no competing interests.
