## [Peer Review File · Nature Communications]

Membrane potential drives the exit from pluripotency and cell fate commitment via calcium and mTORREVIEWER COMMENTS

Reviewer #1 (Remarks to the Author):

This manuscript reports new data on an important emerging field: developmental bioelectricity. The authors show that changes in V_m regulate embryonic cells' exit from pluripotency and germ layer differentiation. They identify KCNH6, a potassium channel, as important for the bioelectric changes that control downstream events through calcium signaling. Their functional work on KCNH6 depletion and transcriptome analysis provides valuable information on how bioelectric signals control transcriptional events important for differentiation. The data are generally of high quality, and I would be supportive of publishing this in Nature Communications, but there are a few critical things that need to be addressed before it becomes publishable:

- The first paragraph of the Results is not a good introduction to this field. It focuses on one single paper [2], attempting to argue that ion pumps are not asymmetrically expressed because a subsequent report failed to find that asymmetry. The text ignores the fact that [2] also reported physiological asymmetry (direct measurement of proton flux that is different between the 2 sides) at those stages, and subsequent papers (such as <https://journals.biologists.com/dev/article/133/9/1657/52955/Early-H-V-ATPase-dependent-proton-flux-is> and others on K^+ channels) reported both localization and V_m data. Also, several subsequent papers (e.g., from Peter Nemes' group) show numerous other proteins already consistently asymmetric at those stages (which contradicts the authors' claims on p. 4 that the first steps of LR symmetry breakage take place in the paraxial LRO). However, the much more important thing is that this debate is not relevant here and does not introduce the topic in a useful manner: what readers should be seeing instead, at the beginning of a paper on V_m 's role in differentiation, is a brief summary of what is known about V_m and differentiation per se. There are many papers on this already, and a few reviews. This is what should be briefly overviewed; otherwise, the reader gets an oddly stilted view of the field, as if nothing had been done on the question of V_m and differentiation and organogenesis more broadly. It ignores key existing data in the field; this needs to be remedied, in favor of the material on the H-K ATPase.

- Related to this, subsequent claims in the text about results showing that V_m is essential for LR patterning and that V_m varies during embryonic development fail to cite prior papers that have already shown this, and in particular have published actual maps of voltage change in the early frog embryo (as well as the fact that V_m gradients have been manipulated by molecular-genetic (not just pharmacological) means to both cause and repair various types of developmental defects). The paper needs to be much more clear about what's new here, and what repeats pieces of what is already known, and cite prior literature appropriately.

- the molecular knockdown and subsequent analyses are the strongest part of the paper. However, the gain-of-function with valinomycin is not really definitive. Especially in Nature Communications, a molecular reagent (e.g., an ion channel mRNA misexpression) should be expected for this kind of data, in a model system in which misexpressing channels is very easy. It especially would allow spatial targeting, which would add value to this report (and is not derivable from any of the measurements, drug data, or knockdown results). Specifically, the claim that "LR patterning defects can be traced to abnormal formation of the paraxial mesoderm of the LRO" is not proven here - it's possible (as prior papers have shown) to get LR defects, including Pitx etc. gene sidedness defects, by targeting mechanisms functioning outside of the LRO. To really make this claim, they should misexpress a channel in precursors, or outside of, the LRO (confirming targeting), and see which ones (or both) cause the defects. Again, I'm not sure there's any need to get into the LR debate in a paper titled "Membrane potential drives the exit from pluripotency and the ontogeny of cell fate via calcium and mTOR", but if the authors do want to make claims like this, then they need to be supported by data and take account of the rich literature in this field that has addressed many aspects of this question. As it is, readers will get a very stilted, piecemeal view of that question.

- calcium: the authors make conclusions about calcium *levels*, but I think more needs to be said about whether it is the overall level, spatial pattern, or temporal pattern of calcium that is relevant here. A number of studies have already pointed to calcium flux as one transduction mechanism for Vm; what should readers take away here as far as what aspect of calcium (total amount, frequency, localization, etc.) is actually important? Also, somewhere it needs to be said that while calcium certainly seems implicated here, these data do not rule out the possible additional involvement of other known transducers of Vmem into transcriptional and morphogenetic events.

- "However, our study indicates that it is Vm and not K+ itself that is critical" - exactly what data prove this? I think it's very likely to be true, but in many prior studies, the way to prove that has been to get to the same Vm change (and the same phenotype) via a different ion, for example sodium or chloride. This is the gold standard in the field for being able to claim Vm vs. specific ion. If there are other data suggesting this that I have missed in this paper, it should be specifically mentioned when this claim is made on p. 484. But overall, the better thing to do is to do an experiment with chloride channel modulation to depolarize/hyperpolarize and see if the same effects are seen. This has been published to work well in Xenopus.

- figure 2: how are "normal" and "abnormal" quantified, for gene expression patterns?

- p. 6 (line 232-) is the best part of the paper. Really nice data here.

- line 335-: there have already been publications reporting high-throughput analyses of gene expression changes downstream of depolarized frog embryos; the timed analysis is a very nice addition to that literature, and that could be emphasized explicitly (and data compared to the existing findings).

- line 429- - whether findings apply to human cells. This approach has already been done in human mesenchymal stem cells - none of those findings are mentioned in the text. There are novel data here - I am not suggesting that the content isn't new or valuable, but currently it's presented in a vacuum, as if doing all this for the very first time, missing the opportunity to educate readers about the field and to compare these data with what is already known. Prior literature addressing these topics has to be dealt with. For example, in mouse, from Calegari's lab:
<http://www.ncbi.nlm.nih.gov/pubmed/21126173>

- "ion channels in structural heart disease was less clear and controversial. Our work provides a plausible mechanism for CHD pathogenesis of ion channels" - again, this reads as if this paper settles a controversy in a definitive way, over what existed. That is not a fair picture of the state of knowledge. There are several papers on for example HCN channels in structural cardiac defects, and there was no controversy - specific mechanisms, including effects on transcription, have been shown clearly. Also, papers on human channelopathies (and other model systems) implicate channel malfunction in structural heart disease.

- Vm is known to fluctuate with the cell cycle (reviewed in <http://www.ncbi.nlm.nih.gov/pubmed/23024660> and many other papers). Do the authors know at what point of the cell cycle their measurements were taken? Are there interactions between cell cycle control and differentiation? There is no mention of this issue that I can find in the text.

Reviewer #2 (Remarks to the Author):

This is a very good and straightforward paper that provides the unexpected insight that membrane polarity is an important determinant of the ability to differentiate. Several methods that inhibit, or

mutate KCNH6 lead to depolarization, and prevent differentiation. Thus the normal role of KCNH6 is to allow efflux of K⁺ ions, causing increased intracellular negative charge (hyperpolarization). This effect is mimicked by increasing extracellular K⁺, which also reduces K⁺ efflux from the cell. The effects can be reversed by increasing efflux with valinomycin, so the effects are ionic ones, rather than some other specific function of KCNH6.

Depolarization also increases the activity of voltage gated Calcium channels, increasing calcium concentration in embryonic cells.

Assays for candidate gene transcripts shows a loss of differentiation, and persistent expression of pluripotency genes, and persistent extension of pluripotency in the response of ectodermal explants to activin. Using a high temporal resolution RNA seq analysis, the authors find that there is an initial burst of Calcium induced genes in depolarized embryos, including mTOR targets, followed by the wave of pluripotency transcripts. Interestingly, the mTOR group of targets are subject to regulation by phosphorylation, and in turn by Calcium signaling. For example, Ets1 phosphorylation is induced by Calcium signals, and this inhibits Ets1 binding activity. The case for this level of regulation is not direct, and is argued based largely on motif analysis. I find the argument compelling, but it would of course be stronger if there were something more direct. However, I don't think this is a necessity for acceptance.

Inline with the arguments on pluripotency factors, the next wave of gene activity correlates with the presence of FoxH1, Sox, and POU motifs, which promote the early pluripotent phase of gene expression

The authors examine differentiation more generally by using human pluripotent cells, and make similar observations, that hyperpolarization prevents differentiation, while depolarization.

The only aspect of the paper I find odd is the apparent rewriting of the logic of the discovery. This group is known for its excellent work tracing the underlying genetic causes of heterotaxy in sporadic cases. This has led to a number of pathways and genes involved in left right organizer signaling. I am therefore unconvinced that they started with questions on electricity, and then turned to CHD patients to solve them. Surely it was the other way around! Perhaps they consider that this makes a logical story, but it doesn't to me.

Overall, the paper is fascinating, and important in providing a well documented case of membrane potential's effects on differentiation. Further to this discovery, it sorts out yet more of the oddball electrical claims on Left right patterning made by the Levin group, which are steadily being refuted. The paper therefore makes important contributions to the fields of Bioelectricity, and on general developmental processes and pluripotency.

When there are directions of hyper and depolarizing effects, it is easy to lose track of what direction things are going in, so if the authors do any rewriting, I would encourage them to include the directionality of signaling and effect in their explanations, and avoid sentences which do not, such as "Here, we show that changes in membrane voltage (Vm) regulate the exit from pluripotency and the onset of germ layer differentiation in the embryo"

Reviewer #3 (Remarks to the Author):

The manuscript, titled 'Membrane potential drives the exit from pluripotency and the ontogeny of cell fate via calcium and mTOR' by Sempou et al., demonstrated that membrane potential plays an important role in embryonic development, specifically affecting the gastrulation process. The majority of the work was performed in xenopus embryos, showing that depolarization of early embryonic cells increases the uptake of calcium into cells at the gastrulation stage, which leads to a fault in the ability of the cells to exit pluripotency and differentiate into paraxial mesoderm. To further determine the

molecular mechanisms, authors also performed high-resolution temporal transcriptome analysis and discovered an upregulated expression of genes associated with mTOR signaling in depolarized embryonic cells. The work is interesting as there are limited reports that a bioelectric pathway has the ability to regulate pluripotency and cell fate determination.

However, there are several issues that need to be clarified in the manuscript.

1. The *kcnh6* gene manipulation experiments.

a) It is clear that the reason that *kcnh6* was targeted was because KCNH6 defects in humans cause CHD/Htx. However, is the xenopus *kcnh6* an ortholog of human KCNH6? Are any other xenopus *kcnh* genes also orthologs of KCNH6? Although *kcnh6* is indicated in the manuscript as the target, all the primer sequences, including *kcnh6* MO and CRISPR sgRNAs, seem to be associated with *kcnh2* rather than *kcnh6*. Therefore, it would be helpful if the authors could provide the accession number for the gene.

b) It lacks direct evidence that the targeted gene has indeed been disrupted. Is there an antibody that could be used to verify this? If there is no available antibody, WMISH may be used as some genes may exhibit mRNA decay after being targeted by CRISPR. In addition, were the control embryos injected with Cas9 proteins?

c) Does '*kcnh6*'-MO affect mCherry expression? mCherry reveals much lower expression in '*kcnh6*'-MO cells than the controls, and more nuclear localization in the '*kcnh6*'-MO cells (Fig. 1o).

2. The treatment of the Rapamycin experiment. In Fig 3f-h, they show that Rapamycin reduced the expression of both *pou5f3.3* and *ventx1.2* in *kcnh6* CR and hiK embryos where both genes had abnormally high expression. Given that the treatment was carried out for several hours at a high concentration (20 µg/ml), Rapamycin may have affected the overall syntheses of nucleotides and proteins. Thus, it would be better to provide evidence to show that the reduction was specific and that non-relevant genes are not affected.

3. Experiments on hESCs.

a) It is difficult to tell the difference on staining intensity between each of the treatment groups. It would be better to perform immunoblotting to verify the results quantitatively.

b) It is unclear how long the cells were treated with Rapamycin and at what concentration (Fig. 4c-d). It seems that the treatment is toxic to the cells as not many cells are left after the treatment, which itself could lead to differentiation. Thus, it is difficult to tell whether the low expression of the pluripotent markers is due to a toxic effect or mTOR inhibition.

c) As the authors pointed, inhibition of mTOR in mouse embryos resulted in the embryos entering diapause, in which the ICM retained their pluripotent status (Bulut-Karslioglu, A. et al., Nature 2016). In the current study, increased mTOR by cell depolarization appeared to delay the exit of embryos from pluripotency, while treatment of Rapamycin promotes differentiation. These two findings seem different. Thus, what do authors think the role of mTOR in the regulation of pluripotency?

Minor points:

1. P9, line 30: 'Xenopus animal cap cells are multipotent'. Are they pluripotent or multipotent? Please clarify.

2. Figure 4: The legend does not match the figure.

We thank the reviewers for their enthusiasm about our work. We have made multiple changes to the text and added additional data to address your concerns. Please note that changes to the text are highlighted in red throughout. Again, we are grateful for your efforts that have dramatically improved the strength of our paper.

Reviewer 1

The first paragraph of the Results is not a good introduction to this field. ... what readers should be seeing instead, at the beginning of a paper on V_m's role in differentiation, is a brief summary of what is known about V_m and differentiation per se. There are many papers on this already, and a few reviews. This is what should be briefly overviewed; otherwise, the reader gets an oddly stilted view of the field, as if nothing had been done on the question of V_m and differentiation and organogenesis more broadly. It ignores key existing data in the field; this needs to be remedied, in favor of the material on the H-K ATPase.

We appreciate the concerns of the reviewer. Our initial draft of the manuscript was exceedingly brief, and we have added introductory material to this revision to highlight what has been described in the field. In addition to LR development as well as what has been described in excitable tissues like muscle cells and neurons, we have included examples of V_m regulation in *Drosophila* wing development as well as chondrogenesis in both facial and limb development.

Related to this, subsequent claims in the text about results showing that V_m is essential for LR patterning and that V_m varies during embryonic development fail to cite prior papers that have already shown this, and in particular have published actual maps of voltage change in the early frog embryo (as well as the fact that V_m gradients have been manipulated by molecular-genetic (not just pharmacological) means to both cause and repair various types of developmental defects). The paper needs to be much more clear about what's new here, and what repeats pieces of what is already known, and cite prior literature appropriately.

We now cite this literature in the revision and highlight these changes in the text. As the reviewer states, there is previous work on V_m in embryonic development. Our work does identify a molecular pathway from potassium channels, to calcium, to mTOR, that includes multiple novel aspects compared to what is available in the literature.

*the molecular knockdown and subsequent analyses are the strongest part of the paper. However, the gain-of-function with valinomycin is not really definitive. Especially in *Nature Communications*, a molecular reagent (e.g., an ion channel mRNA misexpression) should be expected for this kind of data, in a model system in which misexpressing channels is very easy. It especially would allow spatial targeting, which would add value to this report (and is not derivable from any of the measurements, drug data, or knockdown results). Specifically, the claim that "LR patterning defects can be traced to abnormal formation of the paraxial mesoderm of the LRO" is not proven here - it's possible (as prior papers have shown) to get LR defects, including Pitx etc. gene sidedness defects, by targeting mechanisms functioning outside of the LRO. To really make this claim, they should misexpress a channel in precursors, or outside of, the LRO (confirming targeting), and see which ones (or both) cause the defects. Again, I'm not sure there's any need to get into the LR debate in a paper titled "Membrane potential drives the exit from pluripotency and the ontogeny of cell fate via calcium and mTOR", but if the authors do want to make claims like this, then they need to be supported by data and take account of the rich literature in this field that has addressed many aspects of this question. As it is, readers will get a very stilted, piecemeal view of that question.*

We address the reviewer's comment point-by-point:

- 1) First, we thank the reviewer for commenting on the strengths of the paper.
- 2) We regret that the valinomycin experiment may have been confusing. The goal of the valinomycin experiments was to address whether KCNH6 has a role in embryo patterning outside of its role in potassium conductance. Ion channels have been known to have regulatory roles outside of conductance because of protein sequences that extend beyond the transmembrane subunits that form a pore. Valinomycin creates potassium conductance by creating a pore in the membrane due to its chemical structure. However, it cannot participate in any sort of signaling like other ion channels

might. Therefore, valinomycin rescue of the KCNH6 depletion phenotype indicates that potassium conductance is relevant rather than some other signaling function of KCNH6 which could not be answered conclusively by misexpressing an ion channel. We re-state this as “emphasizing the importance of K⁺ flux rather than a specific need for Kcnh6 itself or an alternative role for Kcnh6 in cell signaling.”

- 3) With regard to the claim that “LR patterning defects can be traced to abnormal formation of the paraxial mesoderm of the LRO,” this is based on the following data:
 - a. There is an extremely deep literature that the LRO, the posterior mesoderm that is flanked by endoderm towards the end of gastrulation, plays a critical role in breaking LR symmetry. This has been demonstrated in hundreds (if not thousands) of papers that implicate cilia and nodal signaling in the LRO as critical for breaking symmetry that is then transmitted to the lateral plate mesoderm and marked by Pitx2c. While the details of the mechanism are still actively being investigated, the importance of cilia driven flow and nodal signaling at the LRO is solidly established and conserved in fish, frogs, mice, and humans.
 - b. There is a smaller literature that indicates that chickens may break LR symmetry independently of cilia but still at Hensen’s node, which is analogous to the LRO. In either case “a” or “b”, abnormal patterning of the LRO, where nodal signaling is located, leads to defects in LR patterning.
 - c. There is also some literature that suggests that LR patterning may occur outside of the LRO in *Xenopus* as early as the two-cell stage.
 - d. We have not tried overexpressing the channel to see if it might have an effect outside the LRO. We favor gene depletion experiments in order to identify what the channel’s required role is instead of gain of function experiments that reflect what a channel could do if expressed ectopically.
 - e. As the reviewer suggests, this paper begins with LR patterning but the main focus is really early embryonic pluripotency. Therefore, to address the reviewer’s concern, we have softened the language. Instead of “LR patterning defects can be traced to abnormal formation of the paraxial mesoderm of the LRO.” We will simply state that “Therefore, in *kcnh6* depleted embryos, the paraxial LRO is mispatterned, and a defect in this tissue can be detected already at the onset of gastrulation.” In this way, we can then move on to pluripotency aspects of the paper as suggested by the reviewer.

*calcium: the authors make conclusions about calcium *levels*, but I think more needs to be said about whether it is the overall level, spatial pattern, or temporal pattern of calcium that is relevant here. A number of studies have already pointed to calcium flux as one transduction mechanism for V_m; what should readers take away here as far as what aspect of calcium (total amount, frequency, localization, etc.) is actually important? Also, somewhere it needs to be said that while calcium certainly seems implicated here, these data do not rule out the possible additional involvement of other known transducers of V_m into transcriptional and morphogenetic events.*

This is an interesting point as calcium levels can be highly dynamic. First, we should be clear that our data is based on GCAMP6 fluorescence and therefore provides relative levels rather than quantitative levels of calcium. GCaMP6 is a high-quality reagent with high specificity and extremely high signal-to-noise ratio, and in this case, we have used it to identify what are strong qualitative changes in calcium without intending to measure the exact levels. Nevertheless, we also see that the GCAMP6 fluorescence becomes much broader (affecting many more cells) over the surface of the prospective ectoderm. However, as correctly indicated by the reviewer we cannot comment on the “total amount” of calcium. Instead, we strongly shifted our focus on determining what is upstream and downstream of the calcium signaling, which was instrumental in deciphering the mechanism by which V_m ultimately controls

pluripotency. For example, we find that the voltage gated calcium channel *cacna1c* (and not *cacna1g*) is critical for pluripotency and that mTOR signaling is also involved.

With regard to ruling out other known transducers of V_m , we focused on calcium here and demonstrate a required role. Of course, we cannot/did not rule out all other voltage sensitive transducers which may also be acting downstream or in parallel. We have added a comment to the discussion to indicate this. "While we have not eliminated other voltage responsive signaling, our results indicate that calcium is critical in this context."

"However, our study indicates that it is V_m and not K^+ itself that is critical" - exactly what data prove this? I think it's very likely to be true, but in many prior studies, the way to prove that has been to get to the same V_m change (and the same phenotype) via a different ion, for example sodium or chloride. This is the gold standard in the field for being able to claim V_m vs. specific ion. If there are other data suggesting this that I have missed in this paper, it should be specifically mentioned when this claim is made on p. 484. But overall, the better thing to do is to do an experiment with chloride channel modulation to depolarize/hyperpolarize and see if the same effects are seen. This has been published to work well in Xenopus.

This is an excellent point, and we have performed additional experiments to address this. As suggested by the reviewer, we have manipulated another ion – namely sodium. We replaced sodium in the media with choline which also has a positive charge but cannot conduct through sodium channels and is a poor substrate for sodium transporters and exchangers. Reduction of sodium conductance leads to hyperpolarization of the cell. We show that extracellular sodium depletion rescues the KCNH6 depletion phenotype demonstrating that potassium is not critical. As part of these experiments, we show that reducing extracellular sodium does not only rescue the gastrulation phenotype, but also corrects the V_m based on electrode measurements. Therefore, in addition to the other experiments, we conclude that V_m and not K^+ itself is critical. We have added this new embryonic data to Fig 1m and new electrophysiological data to Ext Data Fig 4.

- figure 2: how are "normal" and "abnormal" quantified, for gene expression patterns?

For this figure as well as other figures with whole mount *in situ* hybridization (WISH), we use the WMISH images in each figure as a key (normal to the left and abnormal to the right) and classify the embryos based on those images. Then the numbers in each classification are simply counted and the ratio of abnormal determined. Also, we have added additional description of our determination of normal/abnormal phenotypes in the respective figure legends, wherever necessary.

- line 335-: there have already been publications reporting high-throughput analyses of gene expression changes downstream of depolarized frog embryos; the timed analysis is a very nice addition to that literature, and that could be emphasized explicitly (and data compared to the existing findings).

We are grateful to the reviewer for pointing out these studies. Unfortunately, we have not been able to find other RNAseq datasets from depolarized embryos. If the reviewer could point us to a specific reference, we would be happy to reference this.

- line 429- - whether findings apply to human cells. This approach has already been done in human mesenchymal stem cells - none of those findings are mentioned in the text. There are novel data here - I am not suggesting that the content isn't new or valuable, but currently it's presented in a vacuum, as if doing all this for the very first time, missing the opportunity to

educate readers about the field and to compare these data with what is already known. Prior literature addressing these topics has to be dealt with. For example, in mouse, from Calegari's lab: <http://www.ncbi.nlm.nih.gov/pubmed/21126173>

We are happy to reference this work. We regret that our original manuscript was so brief and have added this reference as well.

- "ion channels in structural heart disease was less clear and controversial. Our work provides a plausible mechanism for CHD pathogenesis of ion channels" - again, this reads as if this paper settles a controversy in a definitive way, over what existed. That is not a fair picture of the state of knowledge. There are several papers on for example HCN channels in structural cardiac defects, and there was no controversy - specific mechanisms, including effects on transcription, have been shown clearly. Also, papers on human channelopathies (and other model systems) implicate channel malfunction in structural heart disease.

We have softened the text to address the reviewer's concerns. In addition, we have referenced the two papers on HCN channels.

- V_m is known to fluctuate with the cell cycle (reviewed in <http://www.ncbi.nlm.nih.gov/pubmed/23024660> and many other papers). Do the authors know at what point of the cell cycle their measurements were taken? Are there interactions between cell cycle control and differentiation? There is no mention of this issue that I can find in the text.

This is an interesting point as proliferation and the cell cycle have associations with changes in pluripotency. Our membrane potential measurements were taken without regard to the cell cycle and represent averages across all phases of the cell cycle. Grossly, when we examine animal caps in which we have depolarized the cells (either by KCNH6 depletion or application of erg toxin or Barium Chloride), we do not appreciate a dramatic change in cell size compared to control embryos (either smaller or larger). These ectodermal cells are undergoing rapid cell divisions without time for cell growth so a change in cell cycle will likely be reflected in a change in cell size. Because we did not appreciate a noticeable difference, we did not pursue cell cycle changes in this context. To be clear, we recognize that a gross examination of cell size certainly does not exclude a more subtle change in cell cycle. This is an interesting question that we intend to pursue in future experiments. In addition, we have added a comment about future studies addressing V_m in cell cycle and pluripotency.

Reviewer 2

The only aspect of the paper I find odd is the apparent rewriting of the logic of the discovery. This group is known for its excellent work tracing the underlying genetic causes of heterotaxy in sporadic cases. This has led to a number of pathways and genes involved in left right organizer signaling. I am therefore unconvinced that they started with questions on electricity, and then turned to CHD patients to solve them. Surely it was the other way around! Perhaps they consider that this makes a logical story, but it doesn't to me. □

We are grateful for the generous comments about our work in general. It is true that we began these studies primarily focused on CHD/Htx patients without regard for the specifics of KCNH6 in particular. Our initial screens of CHD/Htx candidate genes identified KCNH6 which led to the work presented here. However, in discussion with many of our colleagues in embryonic development and especially in LR development, we felt that it was essential to introduce the concept of ion channels in embryonic development (especially outside of muscle or neuron development) before introducing CHD/Htx patients. We understand that Reviewer 1 might feel differently, but in our discussion with many of the leaders in the field of LR development, the role of ion channels has been controversial and the signaling pathways from V_m to markers of LR patterning not well understood. For this reason, we started with an introduction to the field of Ion channels and LR development. Based on comments from Reviewers 1 and 2, we have made a number

of changes: 1) we have added additional references in the Introduction to address the field of ion channels in development (second half of first paragraph) and 2) we have simplified the introduction to KCNH6 as candidate gene for CHD/Htx.

When there are directions of hyper and depolarizing effects, it is easy to lose track of what direction things are going in, so if the authors do any rewriting, I would encourage them to include the directionality of signaling and effect in their explanations, and avoid sentences which do not, such as "Here, we show that changes in membrane voltage (Vm) regulate the exit from pluripotency and the onset of germ layer differentiation in the embryo"

We have altered these statements as suggested by the reviewer. We agree this helps clarify the text.

Reviewer 3

1. The *kcnh6* gene manipulation experiments.

a) It is clear that the reason that *kcnh6* was targeted was because KCNH6 defects in humans cause CHD/Htx. However, is the *xenopus kcnh6* an ortholog of human KCNH6? Are any other *xenopus kcnh* genes also orthologs of KCNH6? Although *kcnh6* is indicated in the manuscript as the target, all the primer sequences, including *kcnh6* MO and CRISPR sgRNAs, seem to be associated with *kcnh2* rather than *kcnh6*. Therefore, it would be helpful if the authors could provide the accession number for the gene.

This is an excellent point and one we are well aware of. As the reviewer points out, the annotation has been incorrect. Currently, in Xenbase, KCNH6 is incorrectly annotated as KCNH2 and vice versa.

KCNH6: NCBI Annotation XP_012808358.2

We are confident in this annotation. Please see attached tree where XP_012808358 clearly clusters with KCNH6 from multiple species. Additionally, synteny between human and *Xenopus* genomes also indicates that the annotation in Xenbase for *kcnh2* and *kcnh6* are reversed. In the human genome, the neighboring genes for KCNH6 are DCAF7, ACE, CYB561 which is also true for XP_012808358.2 (although this is labeled as KCNH2 in Xenbase). Reciprocally, in the human genome, KCNH2 is adjacent to NOS3, ABCB8, SLC4A2, which is adjacent to the incorrectly annotated KCNH6 in Xenbase.

We have contacted Xenbase which has confirmed the misannotation and will make corrections. We have added NCBI Annotation XP_012808358.2 to the methods section.

b) It lacks direct evidence that the targeted gene has indeed been disrupted. Is there an antibody that could be used to verify this? If there is no available antibody, WMISH may be used as some genes may exhibit mRNA decay after being targeted by CRISPR. In addition, were the control embryos injected with Cas9 proteins?

Unfortunately, potassium channels are notoriously expressed at low levels and there are no antibodies that can detect the protein either in the embryo or by western blot. In fact, to detect mRNA transcripts by WMISH, we have to expose embryos to BM Purple for two weeks to see any signal.

To address the specificity and efficacy of our gene depletion, we do perform a number of different tests. First, we show that multiple non-overlapping sgRNAs produce the same phenotype (Fig 1m, Ext Data Fig 1f-h) and that CRISPR does indeed lead to gene modification at a significant rate (Ext Data Fig 3). Additionally, we show that a MO also produces the same phenotype as CRISPR (Fig 1m, Ext Data Fig 2f-h) and that this phenotype can be rescued by human KCNH6 mRNA expression (Fig 1m, Ext Data Fig 2l).

c) Does 'kcnh6'-MO affect mCherry expression? mCherry reveals much lower expression in 'kcnh6'-MO cells than the controls, and more nuclear localization in the 'kcnh6'-MO cells (Fig. 1o).

We regret that we selected an image that was not representative. The mCherry expression is not affected by *kcnh6* depletion. We have selected a more representative image in the revision.

2. The treatment of the Rapamycin experiment. In Fig 3f-h, they show that Rapamycin reduced the expression of both pou5f3.3 and ventx1.2 in kcnh6 CR and hiK embryos where both genes had abnormally high expression. Given that the treatment was carried out for several hours at a high concentration (20 µg/ml), Rapamycin may have affected the overall syntheses of nucleotides and proteins. Thus, it would be better to provide evidence to show that the reduction was specific and that non-relevant genes are not affected.

This is an excellent point, and we have repeated this experiment and included *ectodermis* as a differentiation marker. In the new figure, we demonstrate (as before) that *kcnh6* CR and hiK lead to abnormally high expression of pluripotency markers *pou5f3.3* and *ventx1.2* and that rapamycin reduces this expression. Additionally, we now show that *kcnh6* CR and hiK lead to loss of *ectodermis* which is rescued by the application of rapamycin. Therefore, we conclude that rapamycin is blocking mTOR and rescuing differentiation when KCNH6 or hiK is applied.

3. Experiments on hESCs.

a) It is difficult to tell the difference on staining intensity between each of the treatment groups. It would be better to perform immunoblotting to verify the results quantitatively.

While differences in staining intensity may be difficult to see by eye, we do quantitate the data using RAW data files. To address the reviewer's concerns, we decided to perform qPCR. This has the advantage of quantitative data and allows us to also look at the level of the mRNA. Indeed, we see that Oct4 and Sox2 mRNA transcripts are elevated when human stem cells are treated with Erg toxin.

b) It is unclear how long the cells were treated with Rapamycin and at what concentration (Fig. 4c-d). It seems that the treatment is toxic to the cells as not many cells are left after the treatment, which itself could lead to differentiation. Thus, it is difficult to tell whether the low expression of the pluripotent markers is due to a toxic effect or mTOR inhibition.

To address this concern, we have repeated the experiment but changed the starting cell density and now show the expression of pluripotency markers as a function of initial seeding and as a function of final cell density. Pluripotency markers are reduced in the rapamycin treated cells regardless of whether the final density is higher or lower than in the control condition. This new data has been added to the manuscript.

c) As the authors pointed, inhibition of mTOR in mouse embryos resulted in the embryos entering diapause, in which the ICM retained their pluripotent status (Bulut-Karslioglu, A. et al., Nature 2016). In the current study, increased mTOR by cell depolarization appeared to delay the exit of embryos from pluripotency, while treatment of Rapamycin promotes differentiation. These two findings seem different. Thus, what do authors think the role of mTOR in the regulation of pluripotency?

This is an interesting question, and there are details that are worth exploring in future work. For example, the mTOR inhibitor used in the Bulut-Karslioglu et al paper (INK128 or RapaLink-1) inhibits both mTORC1 and mTORC2 complexes. They comment that rapamycin which only inhibits mTORC1, “only marginally extend blastocyst survival.” We note that in our case rapamycin is highly effective in differentiated cells held pluripotent by membrane depolarization. Clearly, mTOR plays a role in pluripotency and diapause, the precise mechanism needs to be elucidated including the roles of mTORC1/2 that may be playing differential roles. Numerous futures studies will examine these roles in further detail.

1. *Pg, line 30: 'Xenopus animal cap cells are multipotent'. Are they pluripotent or multipotent? Please clarify.*

We regret this error. Animal cap cells are pluripotent and this has been corrected in the text. We were under the mistaken presumption that pluripotent cells must be able to self-renew indefinitely.

2. *Figure 4: The legend does not match the figure.*

We regret this error and have corrected it.

REVIEWERS' COMMENTS

Reviewer #1 (Remarks to the Author):

The paper is significantly improved and can be published as-is, I have no more requests. I congratulate the authors on a very nice and valuable study.

One thing, the authors wrote:

> We are grateful to the reviewer for pointing out these studies. Unfortunately, we have not been able to find other RNAseq datasets from depolarized embryos. If the reviewer could point us to a specific reference, we would be happy to reference this.

the transcriptomic data to which I referred was not RNAseq, but microarray, and they are reported here: <http://onlinelibrary.wiley.com/doi/10.1002/reg2.48/abstract>

Reviewer #2 (Remarks to the Author):

The authors have addressed my comments, and have done an admirable job of addressing the other reviewers' comments.

Reviewer #3 (Remarks to the Author):

In this revision, the authors have now provided some additional data to address the concerns. The revised manuscript sufficiently justifies the conclusions made. Regarding the role mTOR signalling in this process, I agree with the authors that it is an interesting and complex issue. For example, genes associated with both mTOR and autophagy showed enrichment in activation in the RNA-seq, but mTORC1 is well-known for its role in inactivation of autophagy. In addition, although rapamycin is a specific inhibitor of mTORC1, prolonged treatment of rapamycin also shows an inhibition on mTORC2 by affecting its assembly (<https://www.sciencedirect.com/science/article/pii/S1097276506002188>).

There are some minor points:

- Fig. 6e and legend could be improved more: a) where are (i) and (ii) in the figure? b) what are those p values for?
- Ext Data Fig 9e: what Y axis stands for? If it is relative intensity (au) as stated, do they all refer to 1k mTeSR Sox2?

We are grateful to the reviewers for their generous comments. “I congratulate the authors on a very nice and valuable study.” “The authors...have done an admirable job of addressing the other reviewers' comments.” “The revised manuscript sufficiently justifies the conclusions made.” Only the third reviewer had any further comments which we have addressed in this final revision:

Fig. 6e and legend could be improved more: a) where are (i) and (ii) in the figure? b) what are those p values for?
Thanks to the reviewer for their sharp eye. We have eliminated (i) and (ii) from the figure (this was left over from a previous version). Additionally, the p values were placed in the wrong position which is now corrected.

Ext Data Fig 9e: what Y axis stands for? If it is relative intensity (au) as stated, do they all refer to 1k mTeSR Sox2?
This is precisely correct. This is true for all of the panels from c-f and so we have removed relative intensity (au) and instead placed this in the figure legend which we think will reduce the confusion.